



# Sensitivity of hydrological machine learning prediction accuracy to information quantity and quality

Minhyuk Jeung[1], Younggu Her[2], Sang-Soo Baek[3], Kwangsik Yoon[1]

[1] Department of Rural & Biosystems Engineering (Brain Korea 21), Chonnam National University, Gwangju 61186, Republic of Korea
[2] Department of Agricultural and Biological Engineering / Tropical Research and Education Center, University of Florida, Homestead, Florida 33186, USA
[3] Department of Environmental Engineering, Yeungnam University, Gyeongsan 38541, Republic of Korea

*Correspondence to*: Kwangsik Yoon (ksyoon@chonnam.ac.kr)

**Abstract.** Machine learning (ML) is now commonly employed as a tool for hydrological prediction due to recent advances in computing resources and increases in data volume. The prediction accuracy of ML (or data-driven) modeling is known to be improved through training with additional data; however, the improvement mechanism needs to be better understood and documented. This study explores the connection between the amount of information contained in the data used to train an ML model and the model's prediction accuracy. The amount of information was quantified using Shannon's information theory, including marginal and transfer entropy. Three ML models were trained to predict the flow discharge, sediment, total nitrogen, and total phosphorus loads of four watersheds. The amount of information contained in the training data was increased by sequentially adding weather data and the simulation outputs of uncalibrated and/or calibrated mechanistic (or theory-driven) models. The reliability of training data was considered a surrogate of information quality, and accuracy statistics were used to measure the quality (or reliability) of the uncalibrated and calibrated theory-driven modeling outputs to be provided as training data for ML modeling. The results demonstrated that the prediction accuracy of hydrological ML modeling depends on the quality and quantity of information contained in the training data. The use of all types of training data provided the best hydrological ML prediction accuracy. ML models trained only with weather data and calibrated theory-driven modeling outputs could most efficiently improve accuracy in terms of information use. This study thus illustrates how a theory-driven approach can help improve the accuracy of data-driven modeling by providing quality information about a system of interest.

## 1 Introduction

Machine learning (ML) techniques have become commonly employed for hydrological prediction due to the availability of large hydrological data repositories and advances in computing resources and techniques (Sun et al., 2020; Xu and Liang, 2021). Studies have demonstrated that ML techniques can predict hydrological variables as accurately or even better than other statistical methods and mechanistic (or theory-driven) modeling (Panidhapu et al., 2020). The prediction accuracy of





ML modeling is known to increase with the volume of data used to train the models (Jha et al., 2018); as such, the accuracy is expected to improve further as hydrological observations and records accumulate over time. However, it remains unclear how prediction accuracy is associated with the characteristics of training data: can any data added to a training set improve the accuracy?

Information theory has served as a mathematical tool to measure the amount of information contained in data and its transfer to another set of data (Shannon, 1948a; Shannon, 1948b). This tool can help us understand the correlations or dependencies among multiple interconnected data sets (Pechlivanidis et al., 2018), which helps determine whether the training data contains information that could improve the accuracy of the model (Nearing et al., 2020). Shannon's entropy, often called marginal entropy (ME), is one of the most commonly used information theories that can quantify information

content in a set of data (Silva et al., 2017). The concept of transfer entropy (TE) was proposed to measure the amount of information transferred from one variable to another (Schreiber, 2000). Previous studies have employed ME to quantify the amount of information in hydrological datasets (Silva et al., 2017) and TE to qualify the interactions between input and output data in hydrological analyses (Bennett et al., 2019; Konapala et al., 2020). Both ME and TE have great potential as concepts and methods to evaluate the informatic characteristics of training data and their impacts on hydrological ML model

performance.

Data-driven methods, including ML modeling, rely on historical records and estimates from other analyses, while theory-driven approaches employ existing hydrological concepts and knowledge for prediction. Mechanistic modeling can be classified as a theory-driven method even when its parameter calibration has the nature of a data-driven approach. Mechanistic models employ different assumptions, knowledge, and methods to conceptualize a hydrological system of

interest, which is why they provide unique predictions. For example, streamflow hydrographs predicted using Hortonian and Dunne's concepts might be substantially different from each other even after parameter calibration (Loague et al., 2010). Information embedded in hydrological theories and models can help improve the performance of data-driven modeling, and the information is considered in the predictions of mechanistic modeling. Weather records are one of the data sets commonly used to train hydrological ML models (Chen et al., 2020). Previous studies have demonstrated that ML models trained only

with meteorological data provide limited accuracy; this is unsurprising given that hydrological processes are usually complicated by many other factors, including topography, soil, land use and cover, geological features, and management practices (Srinivasan et al., 2010; Srivastava et al., 2020). Hence, mechanistic model predictions can be an alternative source of data for the training of hydrological ML models.

Mechanistic modeling often or always requires parameter calibration to consider the hydrological characteristics of an area

of interest. In a technical sense, parameter calibration is an effort to improve the statistical similarity between observed and predicted variables of interest. The prediction accuracy of mechanistic modeling is usually improved through the calibration process. As a result, the amount and/or quality of information in a relatively accurate prediction may be greater and/or higher than that of information in a relatively inaccurate one. When the prediction accuracy of mechanistic modeling is improved by calibrating its parameters, the calibrated model may have more and/or better-quality information than the uncalibrated





model. Thus, a pair of uncalibrated and calibrated mechanistic models for a watershed can be a useful tool to create training data sets with different amounts and/or qualities of information for hydrological ML modeling.

This study attempted to relate the quantity and quality of information contained in sets of training data to the prediction accuracy of hydrological ML models, with the goal of understanding how to improve the accuracy efficiently. Information quantity and quality were quantified using information theory, including ME and TE statistics. Three different ML
algorithms (or models) were tested in the evaluation. The quantity of information was systematically increased by adding weather records and uncalibrated and calibrated theory-driven (or mechanistic) model outputs to training data sets. This study employed a mechanistic model commonly used to predict flow and water quality to represent a theory-driven approach. The data-driven (i.e., ML) and theory-driven (i.e., mechanistic) modeling approaches were applied to predict the flow, sediment, total nitrogen (TN), and total phosphorus (TP) loads of four watersheds. Then, the implications of the evaluation
results and the limitations of this study were discussed, and the future direction of hydrological ML modeling was suggested based on the findings.

## 2 Methods and Materials

### 2.1 Overall procedure

Three ML models, including Random Forest (RF), Support Vector Machine (SVM), and Artificial Neural Network (ANN),
were employed in this study (Fig. 1). The ML models were first trained using data sets collected from four study watersheds, including weather data observed at weather stations within or close to the watersheds and flow discharge, suspended soils (SS), TN, and TP measured at the outlets of the four study watersheds. The Soil and Water Assessment Tool (SWAT) model was selected to represent a mechanistic (or theory-driven) model. The SWAT model was used to produce additional data sets to which the ML models would be trained. The SWAT models were calibrated to flow (or streamflow discharges), SS, TN,
and TP loads measured at the outlets. Then, the outputs (i.e., flow discharge, SS, TN, and TP loads) of the uncalibrated (i.e., SWAT models with the default parameter values) and calibrated SWAT models were used as additional data sets for the training of the three ML models. Here, we assumed there would be a difference in the quality of information contained in the uncalibrated and calibrated outputs of the SWAT model, and we employed the marginal and transfer entropy based on Shannon's information theory, to quantify the amount and quality of information contained in the training data sets. Finally,
the weather data and uncalibrated and calibrated SWAT model outputs were sequentially fed to the three ML models to investigate how the ML models' prediction accuracy reacts to the information quantity and quality of the training data (Fig. 1 and Table 1). Information use efficiency (IUE) was quantified to evaluate the performance of the three ML models, and the four training data sets (Table 1) based on the amount of prediction accuracy improvement made using a unit increase in entropy.




**Preparing training data sets** **(a)**

**Theory-driven modeling**

Uncalibrated   Calibrated

- Discharge   • Total nitrogen
- Suspended solids   • Total phosphorus

**Meteorological observations**

- Rainfall   • Average relative humidity
- Average temperature   • Total solar radiation
- Average wind speed   • Total evaporation

**Calculating information quantity and quality** **(b)**

**Information quantity (amount)**

The degree of how much information (or uncertainty) is contained in variable

**Information quality (coherence)**

The degree of how much information is transferred to the output data

**Training ML models**

**Clustering methods (RF and SVM)**

**Neural network method (ANN)** **(c)**

**Prediction accuracy evaluation**

- Comparing the prediction accuracy of target variable between three different machine learning algorithms
- Cross validating the information quantity and quality with prediction accuracy according to the applied input dataset

**Figure 1.** Overall procedure to investigate the contribution of information quantity and quality to the prediction accuracy of hydrological machine learning (ML) modeling.





**Table 1.** Combinations of data sets used to train hydrological ML models.

| Training Data Sets | Variables |
| --- | --- |
| WDO | P, AT, WS, RH, SR, E |
| WD+UC | P, AT, WS, RH, SR, E, Q_Uncal[*], SS_load_Uncal[*], TN_load_Uncal[*], TP_load_Uncal[*] |
| WD+C | P, AT, WS, RH, SR, E, Q_Cal[*], SS_load_Cal[*], TN_load_Cal[*], TP_load_Cal[*] |
| All | P, AT, WS, RH, SR, E, Q_Uncal[*], SS_load_Uncal[*], TN_load_Uncal[*], TP_load_Uncal[*], Q_Cal[*], SS_load_Cal[*], TN_load_Cal[*], TP_load_Cal[*] |

[*] P, AT, WS, RH, SR, and E represent precipitation, average temperature, wind speed, relative humidity, solar radiation, and evapotranspiration, respectively. The SWAT outputs, including Q, SS load, TN load, and TP load, were used to train ML models separately depending on the target variables. For example, SWAT's SS load simulation results were used only when predicting SS load using ML models.

## 2.2 Data-driven (or machine learning) models

The RF model is based on a regression tree, but it differs as it does not grow with a single tree but rather an entire forest of numerous trees using the bootstrap aggregating technique or bagging technique to help decrease model variance (Breiman et al., 1984; Breiman, 2001). The RF model is known for its ability to be used when there are more variables than observation data, and it does not result in overfitting due to the pruning process (Diaz-Uriate and de Andrés, 2006). The RF model was also reported to offer excellent performance even when predictive variables are irregular (Diaz-Uriate and de Andrés, 2006). 110    In RF modeling, a decision tree grows by splitting a tree node, and it is pruned by removing tree nodes or sections with relatively low explanatory power compared to others (Hasanipanah et al., 2017).

The SVM model divides a high- or infinite-dimensional space using hyperplanes until all data points are separated (Vapnik, 1995; 1998). The SVM model is known to be able to avoid overfitting and produces highly accurate predictions (Aktan, 2011). The goal of the SVM procedures is to identify the optimal hyperplane separating two classes in the high- 115    dimensional space that maximizes the distance between the two data point groups (Ahmed et al., 2017). SVM modeling transforms training data using the kernel function so that a linear hyperplane can separate the data points in high dimensions. Three kernel functions are commonly used: radial basis function (RBF), linear function, and polynomial function. This study employed the RBF, which is the most widely used kernel function (Tao et al., 2008).

The ANN model has been widely used to solve various modeling problems (Khashei and Bijari, 2010). The structure of 120    the ANN model was inspired by the biological structure of the human brain, which is composed of many interconnected





processing elements called neurons (Tosun et al., 2016). The structure is characterized by a network of three layers: input, hidden, and output. The number of input and hidden layers is determined by the number of input variables and the complexity of the problem (Yilmazkaya et al., 2018). Neurons are a critical parameter used in interconnected processing, which is characterized by weights (Tosun et al., 2016). The weights of individual neurons determine how input values are

transferred to other values on the output nodes. The weights of connections between layers are calculated by the backpropagation process, which calculates the gradient of prediction error with respect to weights (Siddique and Tokhi, 2001).

### 2.3 Data normalization and accuracy evaluation

ML modeling is known to have low learning rates when some types of training data have value ranges substantially
different from those of others (Ioffe and Szegedy, 2015). Data normalization techniques are commonly used to rescale the training data from their original ranges into a common value range so that the ML models can be efficiently and quickly trained. Several data normalization methods are available; linear scaling is one of the most widely used, presumably due to its simplicity and efficacy (Raju et al., 2020; Eq. 1).

$$X'=(x - \min(x))/(\max(x) - \min(x)) \tag{1}$$

where $\chi'$ is the normalized value of the data set (ranges from 0 to 1), and x is an original value.

The prediction accuracy of the three ML models was evaluated using the Kling-Gupta efficiency coefficient (KGE; Gupta et al., 2009). The KGE considers the strength of the correlation between observed and predicted variables while also comparing the variables' biases and variances. Thus, compared to the Nash-Sutcliffe efficiency and the coefficient of determination, the KGE is less sensitive to relatively large values that lead to biases toward such values (Nash and Sutcliffe,
1970; Gupta et al., 2009; Eq. 2).

$$\text{KGE} = 1 - \sqrt{(r-1)^2 + (\frac{\sigma_{sim}}{\sigma_{obs}} - 1)^2 + (\frac{\mu_{sim}}{\mu_{obs}} - 1)^2} \tag{2}$$

where $\sigma_{obs}$ and $\sigma_{sim}$ are the standard deviations of observations and simulation results, respectively, and $\mu_{obs}$ and $\mu_{obs}$ are the averages of observed and simulated variables, respectively.

A KGE of 1 indicates perfect agreement between observations and predictions (Andersson et al., 2017). Knoben et al.
(2019) mathematically demonstrated that the KGE value approaches -0.41 when the predicted (or simulated) values of a variable are equal to the average value of its observations. Thus, a KGE value of -0.41 can be interpreted similarly to an NSE value of 0.00, meaning that the predictions may not be a better than the observed mean (Schaefli and Gupta, 2007). In this study, we assumed that predictions would be acceptable or satisfactory when the differences between observed and simulated averages of a variable (or percentage biases) and the variances of the differences are less than 25% for flow, 55%
for SS, and 70% for TN/TP (Moriasi et al., 2007), which correspond to KGEs of 0.54, 0.17, and -0.03 for flow, SS, and TN/TP, respectively, with an arbitrarily selected threshold correlation of 0.30.





## 2.4 Theory-driven (or mechanistic) model

The SWAT model was designed to predict watershed processes based on theories and known mechanisms that control the generation and transport processes of water, sediment, and nutrients (Nietsch et al., 2002). The SWAT model is popularly used to predict water and nutrient loadings at the watershed and basin scales due to its proven applicability to a variety of landscapes and climate zones as well as its simple but defendable modeling strategies. Moreover, the SWAT model can consider various management practices, including application rates and timing of fertilizers and herbicides/pesticides; tillage and low-impact development practices; and agricultural conservation practices such as filter strips, nutrient management plans, terraces, and tile drainage (Her et al., 2017; Her and Jeong, 2018; Li et al., 2021a). Several studies have attempted to improve the prediction accuracy of SWAT modeling by coupling it with ML techniques, for example, to predict peak flow (Senent-Aparicio et al., 2019), water quality (Noori et al., 2020), and aquifer vulnerability (Jang et al., 2020).

Two versions of the SWAT model, namely uncalibrated and calibrated mechanistic modeling outputs, were prepared to generate two sets of training data for the ML models. The agricultural management practices that were compiled from the study watersheds were incorporated into both models (RDA, 2014). The values of all parameters of the uncalibrated SWAT model remained unchanged; thus, the uncalibrated SWAT models do not necessarily represent the hydrological processes of the study watersheds, and they are not likely to reproduce the observed flow, SS, TN, and TP at an acceptable accuracy level. Accordingly, the quality of information contained in the outputs of the uncalibrated SWAT models may be relatively low compared to that of the calibrated SWAT models. The parameter values of the SWAT models were calibrated to flow, SS, TN, and TP observations made at the study watersheds' outlets. The flow, SS, TN, and TP loads predicted using the calibrated SWAT models were assumed to have relatively high-quality information compared to those of the uncalibrated SWAT model. The quantity and quality of information were quantified using the marginal and transfer entropies described in the following section.

The SUFI-2 algorithm, widely used for SWAT model calibration, was used to explore the multi-dimensional parameter spaces of the SWAT models and locate a solution (or a parameter set) close to the global optimum in this study (Sao et al., 2020). The simulation period was split into three: a warm-up period from January 1, 2008, to July 11, 2013; a calibration period from July 12, 2013, to December 31, 2015; and a validation period from January 1, 2016, to December 31, 2017. The types and value ranges of the calibration parameters were determined based on the previous SWAT modeling experience, the understanding of the calibration parameters, and the literature (Tobin and Bennett, 2017; Tang et al., 2021).

## 2.5 Marginal and transfer entropy

This study measured the quantity and quality of information contained in the training data using ME and TE. In general, a data set that is spread out has relatively high entropy, while another data set that is concentrated on a small range of values has relatively small entropy. The ME is defined as the information content of a variable and used to calculate randomness in time series using Eq. 3 (Shannon, 1948; Cover and Thomas, 2006; Silva et al., 2017):





$$H(X) = -\sum_{i=1}^{n} p(x_i) \log_2 P(x_i) \tag{3}$$

where $H(X)$ is a measure of information of a discrete random variable $X$, and $P(x)$ is the probability mass function of variable $x$ in the $i^{\text{th}}$ step.

While the amount of information contained in a variable can be calculated using the ME, we can also calculate the amount of information shared between two variables based on mutual information theory using Eq. 4 (Cover and Thomas, 2006):

$$I(X;Y) = H(X) + H(Y) - H(X,Y) \tag{4}$$

where $I(X,Y)$ is the quantified value between $X$ and $Y$. The mutual information $I(X,Y)$ represents the expected information gained in $Y$ from measuring $X$, or vice versa. From these definitions, we can calculate the conditional entropy by subtracting the amount of information shared between $X$ and $Y$ from $H(X)$, which indicates how much information remains about the entire time series $X$ in case we already know the information content of $Y$.

$$H(X|Y) = H(X) - I(X;Y) \tag{5}$$

These quantities are all symmetrical and do not explain the amount of information exchanged between variables (Bennett et al., 2019). The TE was devised to consider the asymmetric transfer of information between any two-time series $X$ and $Y$ (the information flow from one to another variable), and can be defined as conditional mutual information (Schreiber, 2000):

$$T_{X \to Y} = I(Y_t; X_t | Y_t) \tag{6}$$

where $T_{X \to Y}$ is the transfer entropy from $X$ to $Y$, and $X_t$ or $Y_t$ denotes the variables $X$ and $Y$ in time $t$. Once the ME and TE
were calculated for the modeling experiments with the unique combinations of the ML models and the training data sets (Fig. 1 and Table 1), the prediction accuracy gain was divided by the increases in the quantity (ME) and quality (TE) of information contained in the training data to calculate information use efficiency (IUE):

$$IUE_{ME} = \frac{P_{WD \to ID}}{\sum H(x_{WD \to x_{ID}})} \tag{7}$$

$$IUE_{TE} = \frac{P_{WD \to ID}}{\sum T_{WD_i \to Y} \to \sum T_{ID_i \to Y}} \tag{8}$$

where $P_{WD \to ID}$ is the prediction accuracy gain or increase from using additional straining data sets, as compared to the case of only using weather data for the training. The $H(x_{WD \to x_{ID}})$ and $\sum T_{WD_i \to Y} \to \sum T_{ID_i \to Y}$ denotes the marginal entropy and transfer entropy gain or increase from using additional straining data sets, compared to the case of only using weather data for the training.

## 2.6 Study watersheds and training data acquisition

The Pung-Yeong-Jung (PYJ) river watershed was selected for the modeling experiment of this study. The PYJ watershed can be divided into three sub-watersheds from upstream to downstream: the Wall-Jeong (WJ), Ha-Nam (HN), and Jang-Su (JS) watersheds (Fig. 2 and Table S1). The WJ watershed is nested by the HN watershed, and the HN and JS watersheds are nested by the PYJ watershed. Thus, all direct runoff drained from the three nested watersheds passes the outlet (35°09'58.87" N, 126°49'08.93" E) of the PYJ watershed. The streamflow, SS, TN, and TP concentrations were monitored at the outlets of





the four study watersheds for four years and six months, from July 12, 2013, to December 31, 2017. Most of the drainage
areas were covered by agricultural land uses, including upland and rice paddy fields (covering 41% of the JS watershed and
62% of the WJ watershed) and forest. Urbanized areas cover 5% (WJ watershed) to 31% (JS watershed) of the watersheds.

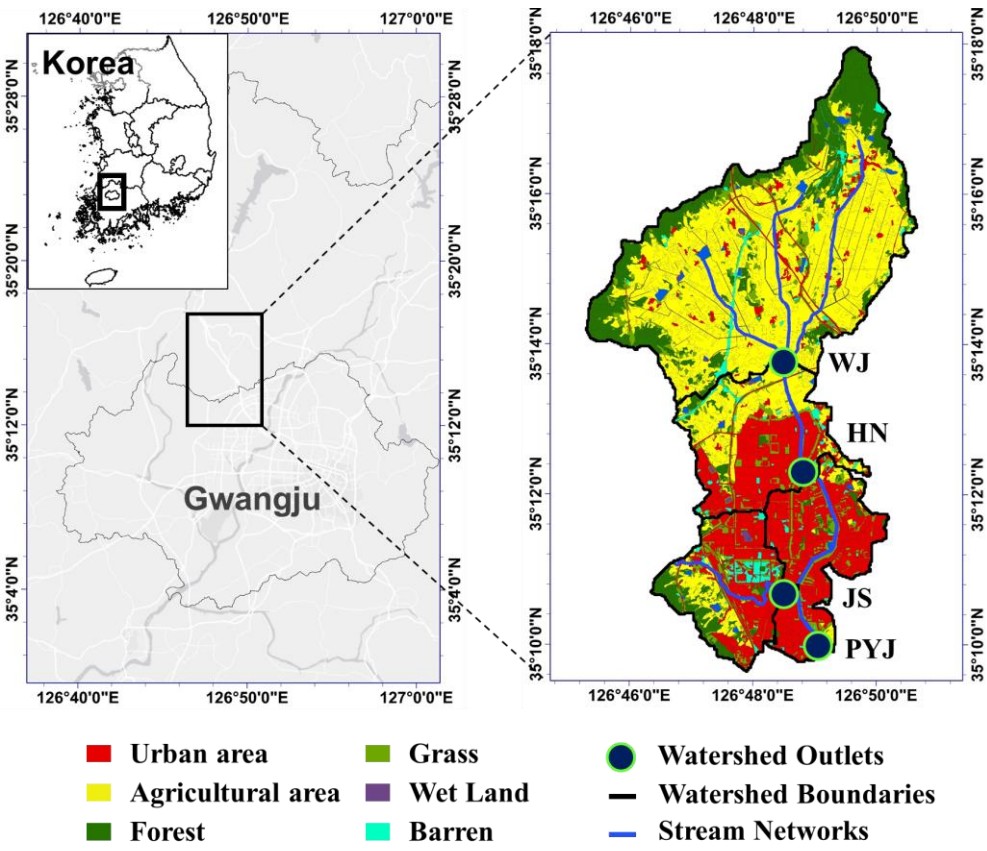

**Figure 2.** Location of the study watersheds and their land uses and covers.

The Korean Meteorological Administration monitors weather variables, including daily AT, P, E, WS, RH, and SR, at a
weather station located approximately 7 km away from the study watersheds. Water pressure sensors and data loggers (OTT
Orpheus Mini, Germany) were deployed at the monitoring sites close to the watershed outlets. The cross-sections of the
streams were surveyed at the monitoring sites, and the velocity of streamflow was then measured using a flow meter
(VALEPORT model 002, UK) across the sections to estimate flow discharge rates. Water quality samples were manually
collected every eight days. During a rainfall event, stream water was collected using an automatic sampler (ISCO portable
sampler 6712, USA), and the sampling interval was reduced to 1 hour to catch the expected large variations of flow rates and
the corresponding water quality concentrations for improved observation accuracy. During the monitoring period, a total of
17 large rainfall events were sampled.





## 3 Results

### 3.1 Training data: Weather records and monitoring data

Sets of training data were prepared using the daily weather records, the watershed monitoring data, and SWAT modeling results (uncalibrated and calibrated outputs; Figs. 3 and S1–S4). The watersheds have four seasons, with relatively short

springs and falls. The watersheds are fairly wet in the summer and dry in the spring. For example, the watersheds receive precipitation of 831–1333 mm annually, with more than half (59% on average) of the precipitation occurring in the summer (from June to September). In spring, the stream might dry up due to the small amount of precipitation and warm air. In the case of the PYJ watershed, streamflow discharges can be large, with as much as 2.64 $m^3/s$ on average in summer, but they are limited (e.g., 1.21 $m^3/s$) enough to reveal the bottom of the stream in spring (from March to May).


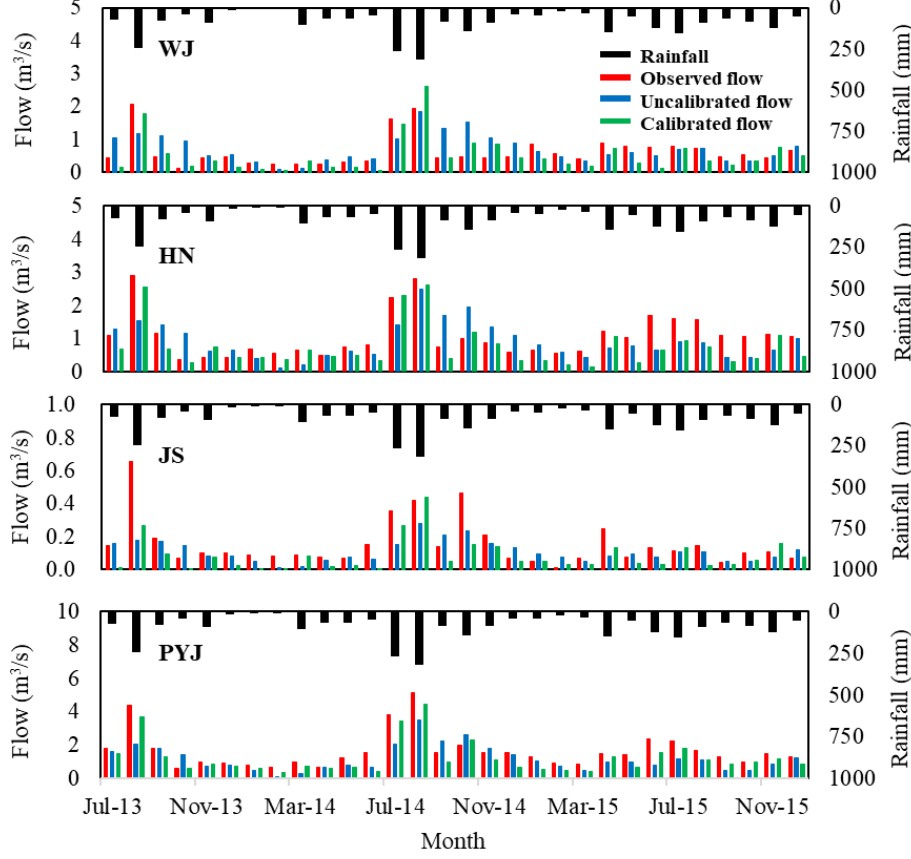

**Figure 3.** Comparison of monthly streamflow predicted using the mechanistic models (i.e., uncalibrated and calibrated SWAT models) and observed during the training period (July 12, 2013, to December 31, 2015). The daily-scale comparisons can be found in the supplementary document (Figs. S1–S4).






The PYJ and JS watersheds had the largest and smallest average daily discharge of 1.69 and 0.16 m³/s, respectively (Table S2). The JS watershed had relatively higher SS concentrations compared to the other watersheds as it includes large construction sites (Mendie, 2005; Pullanikkatil et al., 2015; Adeola-Fashae et al., 2019). In addition, the first flush effects of the urbanized watersheds (e.g., the JS watershed; Table S1) led to higher peak SS, TN, and TP concentrations (Chaudhary et al., 2022). The WJ and HN watersheds had relatively higher TN and TP concentrations, presumably due to agricultural management activities such as fertilizer application and livestock farming in their large agricultural areas (Liu et al., 2012; Table S2).

### 3.2 Training data: Outputs of the mechanistic modeling

The calibrated SWAT model provided acceptable performance in all watersheds (e.g., KGEs equal to or greater than 0.54 for flow, 0.17 for SS, and -0.03 for TN/TP). The average KGE values for all watersheds were 0.68 for flow, 0.45 for SS, 0.40 for TN, and 0.44 for TP (Table 2). However, as expected, the uncalibrated model could not accurately predict the variables; average KGEs for all watersheds were less than 0.41 for flow, 0.02 for SS, -0.20 for TN, and -0.35 for TP. As such, the information quality of the outputs of the calibrated SWAT modeling may be greater than that of the uncalibrated modeling. The quantity and quality of information were evaluated with marginal and transfer entropies.


**Table 2.** Accuracy statistics (KGEs) of a theory-driven (or SWAT) model in the training period. The KGE scores that satisfy the acceptable accuracy criteria (i.e., 0.54 for flow, 0.17 for SS, -0.03 for TN/TP) are in bold.

| Watershed | Flow | | SS | | TN | | TP | |
|---|---|---|---|---|---|---|---|---|
| | Uncal | Cal | Uncal | Cal | Uncal | Cal | Uncal | Cal |
| WJ | 0.49 | **0.71** | **0.28** | **0.52** | -0.28 | **0.41** | -0.39 | **0.43** |
| HN | 0.50 | **0.70** | -0.06 | **0.36** | -0.09 | **0.43** | -0.33 | **0.47** |
| JS | 0.18 | **0.57** | -0.35 | **0.45** | -0.44 | **0.37** | -0.41 | **0.27** |
| PYJ | 0.46 | **0.72** | **0.22** | **0.48** | **0.01** | **0.40** | -0.27 | **0.57** |

### 3.3 Prediction accuracy of machine learning modeling

The four ML models were trained with different sets of training data: weather data only (WDO), the uncalibrated SWAT modeling outputs added to WDO (WD+UC), the calibrated SWAT modeling outputs added to WDO (WD+C), and all training data (All or WD+UC+C). The trained ML models yielded unique performances in the predictions depending on the





training data set types (Fig. 4). Overall, the ML models' flow prediction accuracy consistently improved as additional data sets were added to the training data, including WDO to WD+UC, WDO+C, and All. For example, the WDO case provided acceptable accuracy (KGE of 0.67 greater than the threshold of 0.54) in the prediction of flow using the RF algorithm at the

outlet of the PYJ watershed. When the outputs of the uncalibrated and/or calibrated SWAT modeling were added to the training data, the accuracy of the ML modeling was increased to KGEs of 0.74 (11.6% increase with WD+UC) and 0.91 (37.2% increase with WD+C) in the case of using the RF model. The additional training data sets also improved the accuracy of the water quality ML modeling. However, ML models trained only using the weather data and uncalibrated mechanistic modeling outputs failed to meet the acceptable accuracy levels (i.e., 0.17 for SS and -0.03 for TN/TP; Fig. 4). In Fig. 4, the

KGE scores overall increase from left to right. Negative KGE scores are frequently found in the JS watershed, indicating the models relatively poorly performed for the watershed.

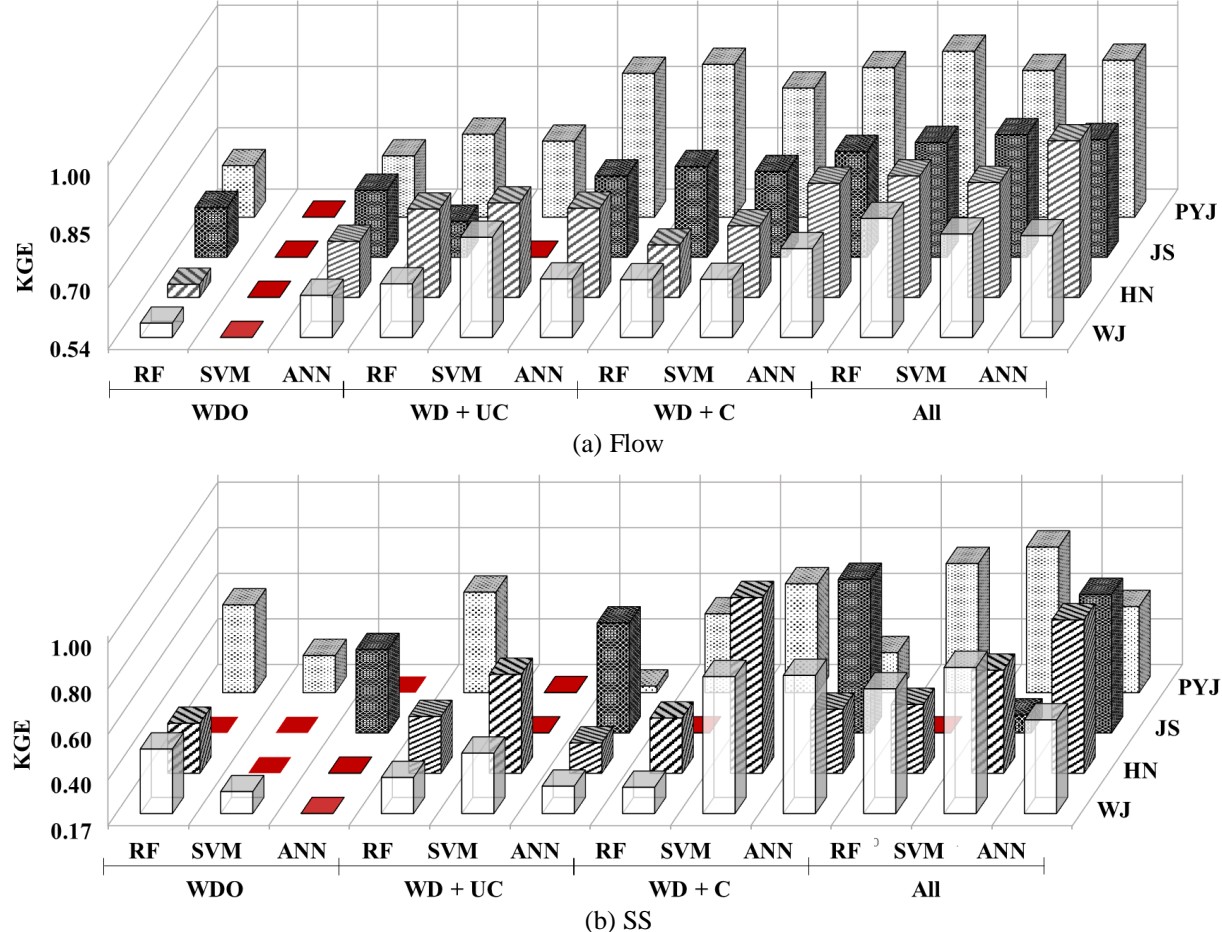

(a) Flow

(b) SS





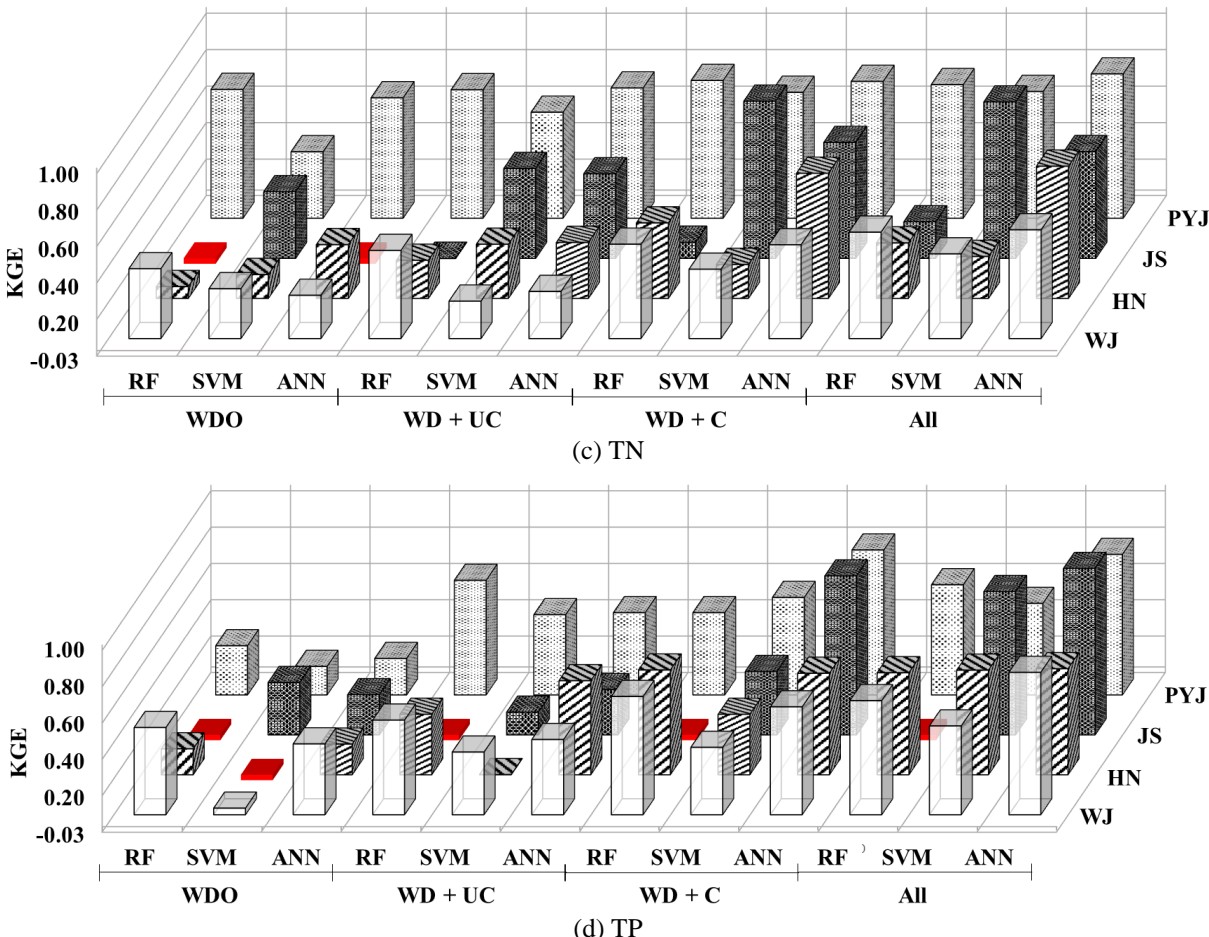

**Figure 4.** Prediction accuracy (KGE) of hydrological ML models trained with the different training data set combinations. The KGE values that do not satisfy the acceptable accuracy levels (e.g., i.e., 0.54 for flow, 0.17 for SS, and -0.03 for TN/TP) are marked with solid red rectangles on the x-y plane.

A flow duration curve (FDC) provides a graphical way of investigating the frequency of extreme events, such as floods and droughts. The FDCs were derived from the observed and predicted flow hydrographs and compared to evaluate prediction accuracy in the frequency domain (Figs. 5 and S5–S7). The FDCs created from flow predictions made using the ML models trained with all training data (the All case) were the closest to the observed FDC in both high (e.g., flooding) and low (e.g., drought) exceedance probability regions. The WDO and WD+UC cases created relatively large differences (under- and over-estimations) between the predicted and observed FDCs, especially for extreme events (i.e., flooding and drought). For example, the differences between the RF predictions and observations for the 5% (flooding) and 95% (drought) exceedance probabilities of the PYJ watershed were 12.1% and 49.9% in the All case respectively, and they increased to 23.0% and 108.6% in the WD+UC case. The findings indicate that the ML models trained with all available training data





sets (the All case) can more accurately predict the extremes than the relatively less trained ML models (the WDO and WD+UC cases).

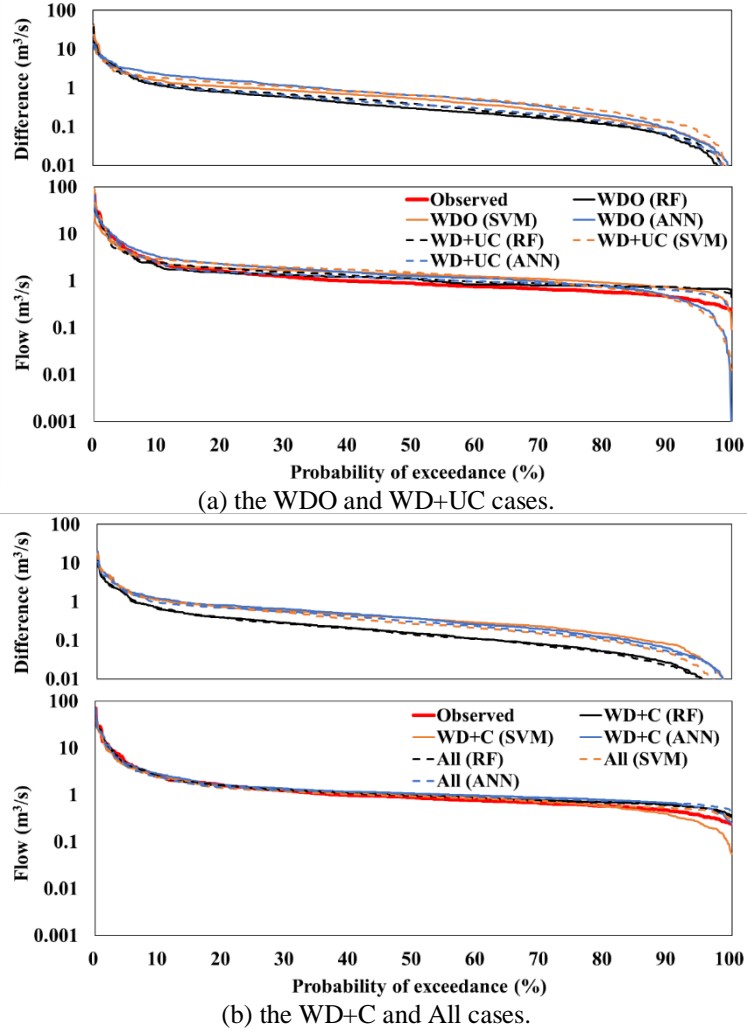

**Figure 5.** Comparison between observed and ML-predicted FDCs at the outlet of the PYJ watershed.

## 3.4 Information quantity and quality

The amount and quality of information contained in the training data sets for the ML training were quantified using the ME and TE concepts, respectively. Then, IUE was calculated to understand how efficiently information quantity and quality can improve the prediction accuracy of hydrological ML modeling. ME of the training data sets generally increased as additional data (i.e., the outputs of the uncalibrated and calibrated mechanistic or SWAT modeling) were added to the weather data (i.e., WDO case; Fig. 6). In the case of predicting the flow of the PYJ watershed, for example, ME increased by



8.7 to 17.9 bits when the uncalibrated and/or calibrated SWAT modeling outputs were added to the training data respectively, compared to the WDO case (Fig. 6[a]). The All cases increased ME more substantially than the WD+UC and WD+C cases, regardless of the watersheds and variables. The WD+C cases did not always increase ME more (or efficiently) than the WD+UC cases, and the ME increases were negligible even when they did occur (e.g., in the case of predicting TP loads);

this confirms that ME does not consider the association between two variables (i.e., watershed responses observed and simulated using the calibrated mechanistic models) in the training data sets, which is one of the features that ME has. Thus, ME does not change depending on the types of ML models as ME only counts information in the training data.

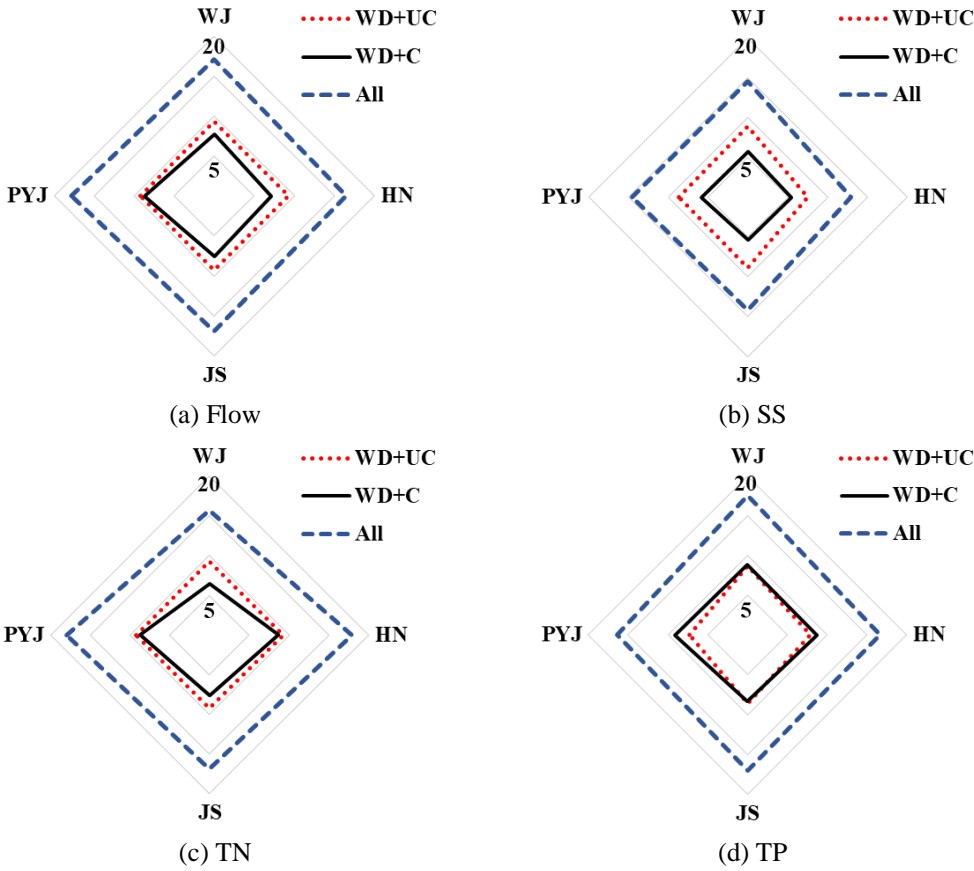

**Figure 6.** Increases in ME due to the addition of additional training data sets. The WDO training data set serves as the
baseline for this comparison.

TE did not always increase with additional training data. For example, in the case of the RF modeling trained with the WDO data set for the WJ watershed, the TE of SS loads decreased from 0.385 to 0.190 and 0.294 when adding uncalibrated





and calibrated mechanistic modeling outputs, respectively (Fig. 7); this indicates that a loss of information was commonly found in the target and training data sets when adding additional data, such as uncalibrated and calibrated modeling outputs, to the training data set. The amount of precipitation information (0.174 bits) was transferred to the SS prediction for the WJ watershed in the case of WDO. However, when adding the uncalibrated mechanistic (i.e., SWAT) modeling output to the training data set, the amount of transferred precipitation information decreased to 0.066 bits, whereas only 0.044 bits were transferred from the uncalibrated SWAT modeling output. Here, the information loss of 0.064 bits can be calculated by subtracting 0.110 bits (amount of information on precipitation and uncalibrated mechanistic modeling output when applying WD+UC training data set) from 0.174 bits. TE considers the amount of information contained in the training data sets and then transfers it into the predictions made using the trained ML models. TE considers the amount of information commonly found in input and output variables and the direction of information flow from variable x to another variable y (Schreiber, 2000). Thus, TE depends on the types of training data sets, prediction variables, and ML models (Figs. 6 vs. 7).

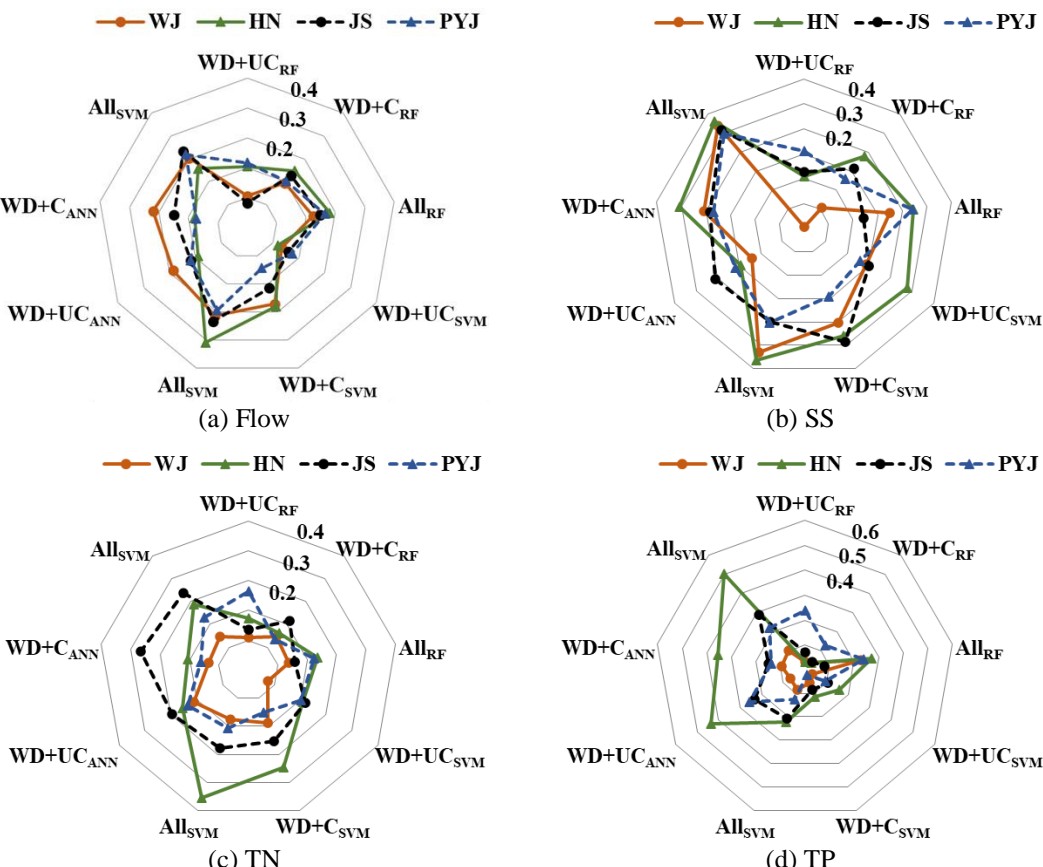

**Figure 7.** Increases in TE due to the addition of training data sets. The WDO training data set serves as the baseline for this comparison.





## 3.5 Information use efficiency

IUE represents the relative improvement of prediction accuracy compared to the baseline per unit change of information
quantity (Eqs. 7 and 8). IUE was calculated by dividing the increases in KGEs (the WDO training data set serves as the
baseline) by the differences between the amount of information quantified using ME (IUE-ME) or TE (IUE-TE) contained in
the training data sets (Fig. 8).

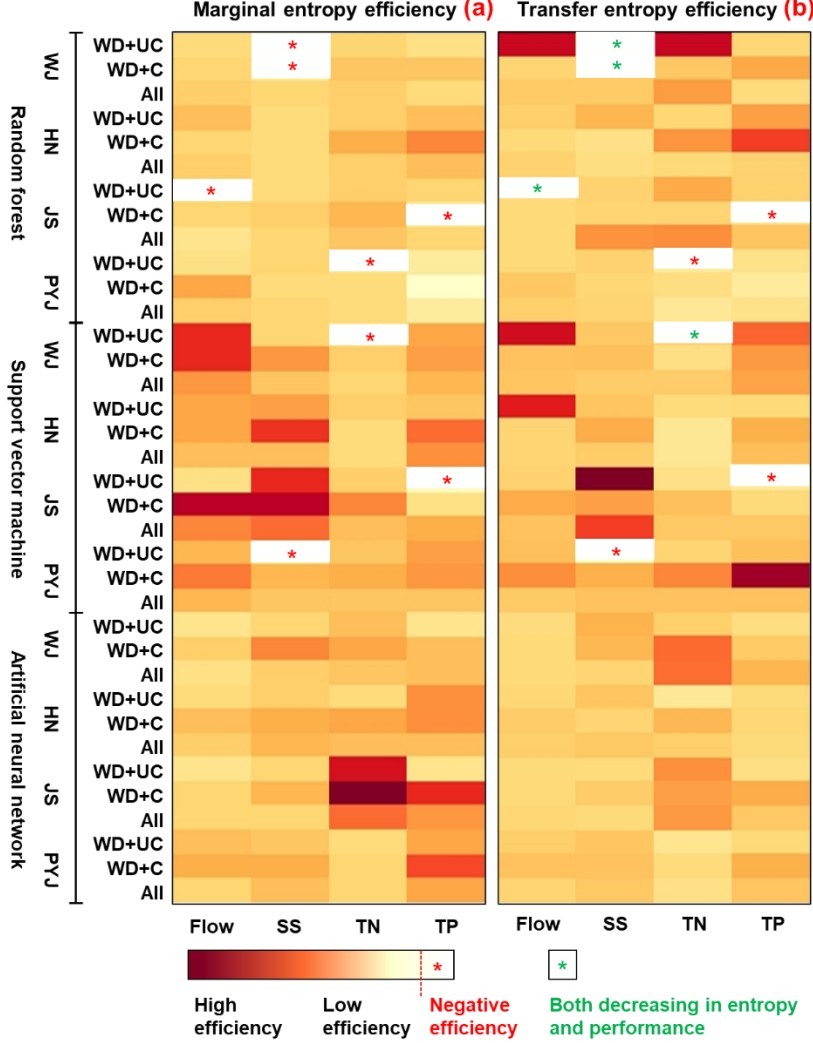

**Figure 8.** Comparison of information use efficiency calculated from the entropy (ME and TE) and accuracy (KGE) statistics
provided by using the different training sets. "Negative efficiency" describes the case where prediction accuracy decreased
with increases in entropy, which is presented with a red symbol. In addition, decreases in both entropy and prediction
accuracy are presented with a green symbol.





The case of WD+C provided a relatively higher IUE-ME compared to the other training data cases (Table S3). This means

that the prediction accuracy of ML modeling can be most efficiently improved when the outputs of the calibrated mechanistic modeling are added to the training data sets (i.e., WD+C). Interestingly, WD+C may be more efficient than the All case, which added the uncalibrated theory-driven modeling outputs to WD+C. This finding implies that information quality can more efficiently improve the prediction accuracy of hydrological ML modeling than information quantity. However, it is worth noting that the All case still provided the best prediction accuracy (or the highest KGE), but its

efficiency in increasing KGE scores was lower than that of WD+C when considering the relative accuracy improvement to the amount of added information.

IUE-ME were often negative, especially in the WD+UC case, indicating that prediction accuracy decreased even when entropy increased (red star in Fig. 8[a]); this is because ME always increases with additional input variables, regardless of their quality or association with the target variables. IUE-TE also showed negative efficiency, which means the KGE

decreased with increases in the TE. Model performance might not necessarily relate to TE because of complicated associations among weather forcings, management practices, watershed feathers, and responses (Konapala et al., 2020). KGEs also decreased when TE decreased (green star in Fig. 8[b]), which implies that TE can capture the decrease in information flow between independent (i.e., weather data, uncalibrated and calibrated modeling outputs) and dependent (or target) variables that may lead to decreased prediction accuracy (or decreased KGEs). This inverse relationship was

primarily detected when adding uncalibrated mechanistic modeling outputs to the training data set, demonstrating the role of information quality in ML modeling training.

## 4 Discussions

This study investigated how the prediction accuracy of hydrological ML modeling is associated with the quantity and quality of information contained in the training data. The results exhibited that prediction accuracy (KGE scores) generally

increased with the amount and quality of information contained in the training data sets (all cases except the cases with stars in Fig. 8). Hence, access to both a large quantity and high-quality information helps increase hydrological ML modeling accuracy. However, the prediction accuracy of hydrological ML modeling and its association with entropy scores were found to be dependent on the study watersheds, target variables, and the ML models.

The accuracy of the ML modeling varied by the watersheds. Regardless of the training data sets, the ML models provided

the best prediction accuracy for the PYJ watershed, which has the largest drainage area, while they did the worst for the JS watershed, which has the smallest drainage area; this implies the potential impact of watershed features and responses (flow, SS, TN, and TP) on ML prediction accuracy. For example, entropy in the watershed responses of the PYJ watershed was consistently higher than that of the JS watershed (Table 3). In the case of WDO, the amount of information contained in the independent variables (i.e., only weather records observed at a single station) of the training data set should be the same for

the PYJ and JS watersheds. However, their responses (dependent or target variables) differ and thus have different entropy





(or information) scores. The responses of the PYJ watershed are spread out over wide value ranges, which means relatively high entropies compared to those of the other watersheds, especially the JS watershed (Figs. 9 and S8–S10). The flow observed at the outlet of the JS watershed are relatively highly biases toward low flow ranges. ML model prediction accuracy was found to be associated with the entropy in the watershed responses (Fig. 10). The results indicated that the

KGE (i.e., prediction accuracy) scores of the ML models generally increased with increases in the amount of information contained in the target variables (Fig. 10).

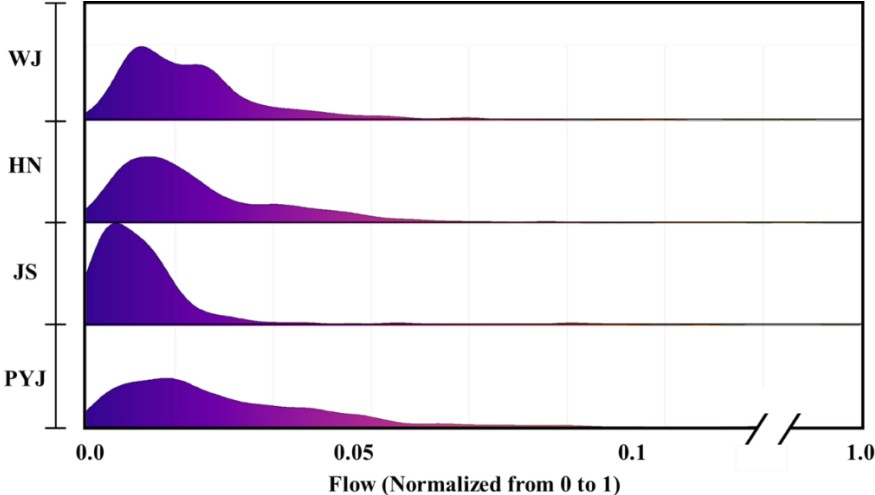

**Figure 9.** Density (or frequency) distributions of observed flow data (i.e., target variable) during the training period. The

flow was normalized from 0 to 1 for each watershed.

**Table 3.** ME quantified for target variables observed at the watershed outlets in the training period.

| Watershed | Flow | SS | TN | TP |
|---|---|---|---|---|
| WJ | 8.680 | 5.507 | 6.080 | 5.757 |
| HN | 8.868 | 5.252 | 5.946 | 5.828 |
| JS | 7.896 | 4.014 | 5.924 | 5.760 |
| PYJ | 8.884 | 5.801 | 6.770 | 6.578 |
| Average | 8.582 | 5.144 | 6.180 | 5.981 |



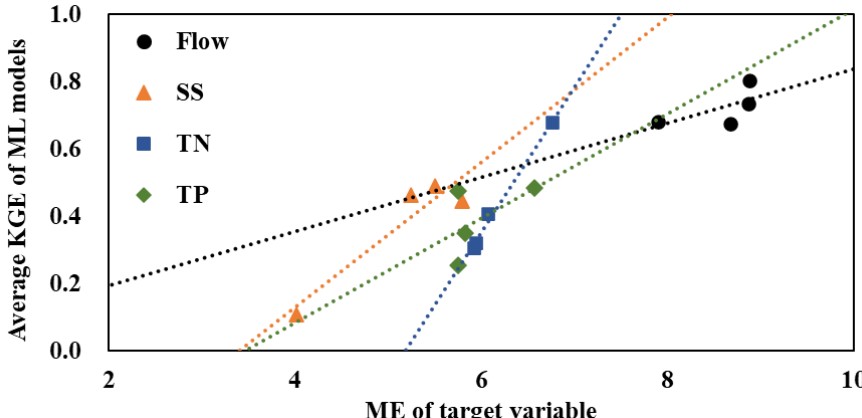

**Figure 10.** Linear relationship between the average KGE scores of three ML models trained using the four training data sets and the ME of the target variables.

The prediction accuracy of the ML models also varied according to the variables of interest (flow, SS, TN, and TP). The RF, SVM, and ANN ML models were best at predicting the flow of the study watersheds compared to the other variables (Fig. 4). For example, the ML models provided KGEs of 0.557 (WDO) to 0.854 (All) when predicting flow versus KGEs of 0.093 (WDO) to 0.607 (All) for SS loads. This variance is presumably because of the previously described differences in the amount of information contained in the watershed responses or target variables, which is also why prediction accuracy varied by watershed. For example, the flow hydrographs commonly have relatively higher entropies (8.582 on average) than the other variables' hydrographs (5.144 for SS, 6.180 for TN, and 5.981 for TP on average) for all study watersheds (Table 3). In the frequency domain, normalized SS load data have the most biased distributions toward low values (small SS loads) and the highest frequencies among the watershed responses or target variables, leading to relatively low entropy in the SS data (Figs. 9 and S8–S10). The SS, TN, and TP concentrations observed at the watershed outlets have relatively small variations compared to the flow (Table S2), which might be attributed to the fact that water quality variables were much less frequently measured (or sampled) than flow (Table S2, the number of observations); thus, potentially large concentration variations might not be apparent in the observations. These comparison results imply that the frequency of water quality sampling can affect the amount of information in training data and the accuracy of hydrological ML model prediction.

None of the ML models consistently provided more accurate predictions than the others (Fig. 4). This finding aligns with other studies that identified no ML model that is universally applicable to all data sets or problems (Alzubi et al., 2018). Some study has demonstrated that the RF model is more accurate compared to the SVM and ANN models (Al-Mukhtar, 2019). Conversely, other study has determined that the SVM or ANN model outperform the RF model (Ahmad et al., 2018). In this study, the RF model provided relatively better accuracy than other ML models when predicting the streamflow of the PYJ watershed using all training data sets (the All case). The ANN model was the only one that could provide acceptable accuracy (KGE of 0.84, which is greater than the threshold of 0.17 for SS) in the prediction of SS loads in the JS watershed





with the WD+C training data. The SVM model provided a relatively greater KGE score than the other models when
predicting the SS loads of the HN watersheds using the WD+UC training data.

The ANN and SVM models could improve their predictions more efficiently in terms of the amount of information (ME) added to the training data compared to the RF model (Fig. 8[a]). The RF model uses a random sampling method to select the feature subspace for each node in growing the trees (Breiman, 2001), which is a model parameter called the "number of variables." Previous studies (Wang and Xia., 2016; Ye et al., 2013) have argued that when applying the random sampling
method to a high-dimension data set, model may select many subspaces that do not include informative features and will increase error bounds for the RF model. This study agrees with previous studies: RF performed relatively poorly when dimension of a training data set was higher (i.e., the large number of independent variables) than SVM and ANN. A traditional ANN model with one or two hidden layers is known to suffer performance degradation due to its rapid growth in the number of connection weights (Krenker et al., 2011). However, one study demonstrated that a deep neural network that
employs numerous hidden layers, such as the one used in this study, could yield promising performance with high-dimensional training data (Liu et al., 2017).

Negative IUE-TE values were found when watershed responses were predicted using the RF and SVM models (red star in Fig. 8[b]), especially in the WD+UC case. The RF and SVM models occasionally failed to utilize additional information contained in the training data, presumably due to the "curse of dimensionality" (Bellman, 1961) and/or the complicated
nonlinear relationship between independent and dependent (or target variables) data. Both RF and SVM models use "piecewise" linear decision boundaries or hyperplanes to partition the input space, but the decision boundaries can be nonlinear overall at the end of the decisions or partitioning. To handle nonlinear cases, SVM models employ a kernel function to transform the nonlinear decision space into a linear one, and RF models use nonlinear decision boundaries (Kirasich et al., 2018). However, studies have supported that nonlinear decision boundaries might not always be able to help
solve the high-dimensionality issue, mainly because random sampling could sample less informative features (or noises) when growing trees (Wang and Xia., 2016; Ye et al., 2013). On the other hand, the kernel function of the SVM model has been found to handle high-dimensional non-linear data well (Huang et al., 2018), which contrasts with our finding. In this study, we used a radial basis kernel function and optimized the ML models using a Bayesian optimization method, which may affect predictive accuracy with feature selection (Shawe-Taylor and Sun, 2011). The ANN model did not provide any
negative IUE scores in this study, and its performance is less affected by the relatively low-quality information included in the training data set (the WD+UC case in Table S3). These findings suggest that the ANN model can more efficiently utilize quality information than the other two models.

## 5 Conclusions

From the modeling experiment, this study revealed that the accuracy of hydrological ML prediction is closely associated
with the quantity and quality of data used to train the models. Prediction accuracy was maximized when all available data



(i.e., the All case) were employed for training, and it was most efficiently improved in terms of information use when relatively high-quality data (i.e., the WD+C case) were added to the training data set. Information use efficiency was affected by the amount of information contained in the dependent (or target) variables of the training data, which varied by watershed and variable type. ML model performance was case-dependent, and the ANN model could more efficiently utilize the quality

information contained in the training data set than the SVM and RF models. Relatively low-quality information (i.e., the WD+UC) case sometimes did not improve prediction accuracy, demonstrating the significance of the quality as well as the quantity of training data. These findings are expected to elucidate the relationship between information and ML modeling accuracy, highlighting the importance of data quality and information in ML model training.

***Author contributions*.** **MJ**: Conceptualization, Software, Validation, Formal analysis, Writing - Original Draft; **YH**: Conceptualization, Methodology, Supervision, Writing - Review & Editing; **SB**: Validation, Formal analysis, Data Curation; **KY**: Conceptualization, Supervision, Writing - Review & Editing.

***Competing interests*.** The contact author has declared that none of the authors has any competing interests


***Acknowledgements*.** This research was supported by a project titled "A Long-term Monitoring for the Nonpoint Sources Discharge" (Yeongsan and Seomjin River Water Management Committee).

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
