# Peer review of "Sensitivity of hydrological machine learning prediction accuracy to information quantity and quality"

_Hydrology and Earth System Sciences, 2024_

## Referee Comment (RC2)

This work provides insights and framework on how to quantify input data quantity, quality, and their impacts on ML predictions for water quality and quantity, with potential to guide future modeling efforts. Here are some main suggestions to improve the manuscript: (1) Expand introduction to include recent work on data synergy and the integration of process-based models and ML. (2). The water quality model seems to exclude nutrient inputs, which weakens the conclusions. (3). Consider simplifying some of the results figures. Not all models, basins, or test cases need to be visualized explicitly at the same time. (4) Consider exploring how the findings can be applied to improve better modeling processes in Discussion.

Detailed comments:

1.Introduction: There have been emerging studies on the "data synergy" effect of data-driven approaches and combining process-based models with ML. However, this manuscript lacks a comprehensive and evaluative literature review. Please consider including some up-to-date work, such as:

- Kratzert, Frederik, Daniel Klotz, Sepp Hochreiter, and Grey S. Nearing. 2021. "A Note on Leveraging Synergy in Multiple Meteorological Data Sets with Deep Learning for Rainfall–runoff Modeling." Hydrology and Earth System Sciences 25 (5): 2685–2703.
- Razavi, Saman, David M. Hannah, Amin Elshorbagy, Sujay Kumar, Lucy Marshall, Dimitri P. Solomatine, Amin Dezfuli, Mojtaba Sadegh, and James Famiglietti. 2022. "Coevolution of Machine Learning and Process-based Modelling to Revolutionize Earth and Environmental Sciences: A Perspective." Hydrological Processes 36 (6). https://doi.org/10.1002/hyp.14596.
- Reichstein, Markus, Gustau Camps-Valls, Bjorn Stevens, Martin Jung, Joachim Denzler, Nuno Carvalhais, and Prabhat. 2019. "Deep Learning and Process Understanding for Data-Driven Earth System Science." Nature 566 (7743): 195–204.

2. Line 80: Can you discuss the rationale for choosing RF, SVM, and ANN as the machine learning models for streamflow predictions to test your hypothesis? Recent studies have suggested that Long Short-Term Memory (LSTM) networks are the state-of-the-art machine learning models for time-series river flow predictions, outperforming other approaches.

3. Table 1: In the current model setup, for both water quantity and water quality predictions, the baseline inputs (WDO) include only climate variables. However, in reality, nutrient inputs (e.g., fertilizers, human waste, and manure) are essential for predicting N and P, regardless of whether using machine learning or process-based models. Do you think it is fair to test TN and TP predictions when WDO includes only climate variables? If nutrient inputs were incorporated, adding loads from SWAT might have less impact. How

much of the conclusions from this study can be applied to water quality ML studies where nutrient inputs are typically included as predictors?

4. Line 255: More information about the SWAT model is needed.

- The manuscript states that SWAT can incorporate management practices, and two of the watersheds are heavily farmed and urbanized. Were inputs and parameters representing agricultural and human processes that significantly impact water quality (TN, TP, and SSD) included and calibrated? Please provide the inputs, parameters, and calibrated values.
- It is also unclear how many SWAT models were developed. Was a single SWAT model used per basin, or was a separate model created for each target variable within a basin?
- Related to (2), can you clarify the calibration process? Was a weighted multi-objective calibration approach applied, or were parameters calibrated for flow first, followed by calibration for water quality?

5. Figure 4: 3D plots are often hard to interpret. Consider using 2D plots for model performance comparison.

6. Figure 5: Could you adjust the figure size to make the duration curves less flat? It's difficult to distinguish the differences. What is the scale of the Y-axis? It appears to be on a log scale

7. Figure 6: All basins and target variables show a similar pattern, with the key takeaway being that ME increases with the amount of data, regardless of data correlation. You might consider leaving one subfigure and moving the rest to the supplemental section, or consolidating them into a single figure. For example, as the conceptual figure suggests, the X-axis could represent different data, the Y-axis ME, and different lines could indicate target variables. Adding uncertainty bands could also capture watershed variance. Just some suggestions to consider

[Figure]

Figure 7: This figure now has too many dimensions—different basins, target variables, model inputs, and ML models—which makes it quite confusing. Could you simplify it to highlight key finds, perhaps by leveraging the suggestions for Figure 6

8. Figure 10: How well are the regressions? Can you report R2 and p-value?

9. Discussion: Several points discussed are common knowledge, making them less novel and somewhat irrelevant. For example: (1) ML model accuracy depends on study watersheds, target variables, and ML model types; and (2) water quality is generally harder to predict than water quantity. The key findings of this work is its quantitative evaluation of data quality and quantity and their relationship to model performance. Could you expand on how these findings can guide future modeling efforts, such as optimizing input selection, implementing quality control measures, or integrating insights with process-based models

10. Maybe I missed it. Did you talk about uncertainty and limitations of your work?

---

## Author Comment (AC1)

**Sensitivity of hydrological machine learning prediction accuracy to information quantity and quality**

**RC1.1**: The manuscript entitled "Sensitivity of hydrological machine learning prediction accuracy to information quantity and quality" present a valuable discussion about the influence of information quantity and quality on the performance of machine-learning-based (ML) models for hydrological prediction.

**Response to RC1.1**: Thank you for recognizing the value of this study.

**RC1.2**: Below are some points regarding its methodology, results, and potential areas for improvement:

It is quite trivial that calibrated models can offer training samples with high quality and thus help machine learning models achieve significant performance improvement. Could you please further clarify which key scientific findings/insights can be offered by this study?

**Response to RC1.2**: We appreciate your feedback and understand your point. Indeed, it is reasonable to expect that calibrated models provide higher-quality training data compared to uncalibrated models. However, our question is: what specifically makes the outputs of a calibrated model high-quality training data? In other words, if calibrated model outputs do improve machine learning model accuracy, why is that the case? The aim of this study is not simply to recommend using calibrated mechanistic model outputs to enhance accuracy. Instead, we seek to understand how to improve the accuracy of hydrological machine learning models efficiently by exploring the underlying qualities of training data that contribute to this improvement. In this study, we examined the relationship between the quantity and quality of information in training datasets and the prediction accuracy of hydrological machine learning models. Our assumption is that training sample quantity and quality can be quantified using information theory measures, specifically marginal entropy for quantity and transfer entropy for quality.

To clarify our approach, we have added our research questions and hypotheses to the article (after the sentence ending in Line 69 on Page 3), providing readers with a clear understanding of the core objectives of this study: "The research question that this study tried to answer was how the quantity and quality of information in training datasets, as measured by marginal entropy and transfer entropy, can affect the prediction accuracy of hydrological machine learning models? Our hypothesis was that a higher quantity and quality of information in training datasets, as indicated by increased marginal entropy and transfer entropy, would positively correlate with improved prediction accuracy in these models."

**RC1.3**: Figure 1. classifies Random Forest (RF), Support Vector Machine (SVM) as clustering methods, Artificial Neural Network (ANN) as neural network method. What are the essential differences between the two categories of ML models and whether such differences will influence the following discussion?

**Response to RC1.3**: The essential differences between clustering methods and neural network methods in machine learning are rooted in their objectives, underlying mechanisms, and applications. The clustering methods are unsupervised learning techniques that aim to group data points into clusters based on similarity or distance measures without prior knowledge of labels. They are primarily used for data exploration, identifying inherent patterns, and segmenting data into meaningful clusters. Neural networks are typically used in supervised or reinforcement learning contexts, aiming to learn complex patterns in labeled data for prediction, classification, or decision-making tasks. They are known to be versatile and well-suited for high-dimensional and non-linear data. These differences have important implications for the choice of machine learning models. We have revised the last paragraph of the discussion section and added another paragraph with a focus on how the distinctions between clustering methods and neural networks influence their respective effectiveness and applications:

"Negative IUE-TE values were observed when watershed responses were predicted using RF and SVM models (red star in Fig. 8[b]), particularly in the WD+UC case, suggesting challenges in leveraging additional information from training data. The RF and SVM models, which rely on "piecewise" linear decision boundaries or hyperplanes to partition input space, struggled to manage the "curse of dimensionality" (Bellman, 1961) and complex non-linear relationships between variables. While SVM models use kernel functions to transform non-linear decision spaces into linear ones, and RF models employ non-linear decision boundaries, prior studies indicate that such methods are not always effective in resolving high-dimensional issues, often sampling less informative features (Wang and Xia, 2016; Ye et al., 2013). Despite the radial basis kernel function and Bayesian optimization employed in this study to enhance SVM performance (Shawe-Taylor and Sun, 2011), the model's predictive accuracy remained inconsistent. Conversely, the ANN model avoided negative IUE scores, demonstrating its resilience and ability to more efficiently utilize quality information, even with lower-quality training data in cases such as WD+UC (Table S3).

Neural networks, particularly the ANN model, excel in handling high-dimensional, non-linear data, making them more effective than RF and SVM for this study's hydrological predictions. With diverse features such as precipitation, temperature, and watershed characteristics contributing to accurate predictions, the ANN model utilized the rich, high-dimensional data from calibrated and uncalibrated SWAT outputs to achieve strong performance. Unlike clustering methods, which primarily group data without a predictive function, neural networks improve prediction accuracy through learning from labeled data and adapting to input quality. The

absence of negative IUE scores for ANN underscores its flexibility and robustness. These findings affirm the ANN model's suitability for high-dimensional, quality-driven hydrological modeling, highlighting its advantage over other methods in tasks requiring predictive precision and adaptability to data complexity."

**RC1.4**: For Sect. 2.2, the input variables of machine learning models are not clear. It might need further explanation about the setting-up process of machine learning models.

**Response to RC1.4**: We appreciate your observation regarding the insufficient explanation of the machine learning model setup process. To address this, we have provided a more detailed description of the input variables and clarified the dataset division process in the revised manuscript. We have also elaborated on the optimization procedures for each model, which were conducted using Bayesian optimization to ensure efficient and accurate parameter tuning.

In the sub-section of "2. Method and Materials: 2.2 Data-driven (or machine learning) models" of the revised manuscript, we state: "The optimization of three machine learning models—RF, SVM, and ANN—was carried out using Bayesian optimization, a method that improves decision-making efficiency by iteratively identifying the most promising hyperparameter configurations (Jones, 2001). Compared to traditional grid or random search methods, Bayesian optimization is notably more efficient in finding optimal hyperparameters (Yu and Zhu, 2020). For the RF model, key parameters such as the maximum number of splits, the number of predictors per split, and the number of trees were optimized. In the case of the SVM model, the kernel scale, epsilon, and cost parameters were fine-tuned. For the ANN model, optimization focused on activation functions and layer sizes. These optimizations were designed to enhance each model's performance by leveraging input variables—including precipitation, temperature, and watershed characteristics—that were carefully selected to align with the study's objectives."

**References**

Jones, D.R., 2001. A Taxonomy of Global Optimization Methods Based on Response Surfaces. Journal of global optimization, 21, pp.345-383.

Yu, T., Zhu, H., 2020. Hyper-parameter optimization: A review of algorithms and applications. arXiv preprint arXiv:2003.05689.

**RC1.5**: Line 151: why is the threshold correlation arbitrarily selected as 0.30?

**Response to RC1.5**: The interpretation of a correlation coefficient of 0.30 varies depending on the context and field. While some define 0.30 as the threshold for a "medium" correlation (Woolf, 2009), others describe it as a low correlation (Asuero et al., 2006) or weak correlation (Schober and Schwarte, 2018). Low or weak correlations suggest some degree of structure or association, as opposed to no meaningful correlation when values fall below this threshold. Since the correlation coefficient measures linear agreement between two variables (e.g., observed vs.

predicted), achieving high linear correlations in hydrological modeling is uncommon due to the inherently complex and nonlinear nature of hydrological processes, especially on relatively fine temporal scales such as daily. Based on the literature, we determined that a correlation value of at least 0.30 represents the minimum acceptable strength of correlation between observed and predicted hydrological variables for this study.

**References**

Woolf, P. J., 2009. Chemical Process Dynamics and Controls. United States, University of Michigan Engineering Controls Group. pp.13.13.1. (https://eng.libretexts.org/Bookshelves/Industrial_and_Systems_Engineering/Chemical_Process_Dynamics_and_Controls_(Woolf)/13%3A_Statistics_and_Probability_Background/13.13%3A_Correlation_and_Mutual_Information)

Asuero, A.G., Sayago, A. and González, A.G., 2006. The correlation coefficient: An overview. Critical reviews in analytical chemistry, 36(1), pp.41-59.

Schober, P., Boer, C. and Schwarte, L.A., 2018. Correlation coefficients: appropriate use and interpretation. Anesthesia & analgesia, 126(5), pp.1763-1768.

**RC1.6**: Figure 4. uses 3D plotting which might make comparison between different cases and models difficult. Could you please use a 2D figure with legends instead?

**Response to RC1.6**: Thank you for your valuable feedback; we understand your concerns and have made the necessary adjustments. The figures have been revised to a 2D format for improved clarity, with each watershed represented by a distinct symbol. Additionally, the training dataset is now plotted along the x-axis for better visualization.

[Figure]

(a) Flow

[Figure]

**Figure 4.** Prediction accuracy (KGE) of hydrological ML models trained with the different training data set combinations. The KGE values that do not satisfy the acceptable accuracy levels (e.g., i.e., 0.54 for flow, 0.17 for SS, and -0.03 for TN/TP) are included in gray areas.

---

## Author Comment (AC2)

**Sensitivity of hydrological machine learning prediction accuracy to information quantity and quality**

**RC2.1**: This work provides insights and framework on how to quantify input data quantity, quality, and their impacts on ML predictions for water quality and quantity, with potential to guide future modeling efforts. Here are some main suggestions to improve the manuscript: (1) Expand introduction to include recent work on data synergy and the integration of process - based models and ML. (2). The water quality model seems to exclude nutrient inputs, which weakens the conclusions. (3). Consider simplifying some of the results figures. Not all models, basins, or test cases need to be visualized explicitly at the same time. (4) Consider exploring how the findings can be applied to improve better modeling processes in Discussion.

**Response to RC2.1**: Thank you for your encouraging comments and constructive suggestions. We have carefully addressed all your feedback in the revision, which we believe has significantly enhanced the manuscript. Below, we provide a detailed explanation of how each comment has been addressed.

**RC2.2**: Introduction: There have been emerging studies on the "data synergy" effect of data - driven approaches and combining process-based models with ML. However, this manuscript lacks a comprehensive and evaluative literature review. Please consider including some up-to-date work, such as:

- Kratzert, Frederik, Daniel Klotz, Sepp Hochreiter, and Grey S. Nearing. 2021. "A Note on Leveraging Synergy in Multiple Meteorological Data Sets with Deep Learning for Rainfall–runoff Modeling." Hydrology and Earth System Sciences 25 (5): 2685–2703.
- Razavi, Saman, David M. Hannah, Amin Elshorbagy, Sujay Kumar, Lucy Marshall, Dimitri P. Solomatine, Amin Dezfuli, Mojtaba Sadegh, and James Famiglietti. 2022. "Coevolution of Machine Learning and Process-based Modelling to Revolutionize Earth and Environmental Sciences: A Perspective." Hydrological Processes 36 (6).
- Reichstein, Markus, Gustau Camps-Valls, Bjorn Stevens, Martin Jung, Joachim Denzler, Nuno Carvalhais, and Prabhat. 2019. "Deep Learning and Process Understanding for Data-Driven Earth System Science." Nature 566 (7743): 195–204.

**Response to RC2.2**: Thank you for sharing the recent studies on the data synergy effect of combining two different modeling approaches in hydrological predictions. As per your suggestion, we have incorporated the recommended literature into the introduction, presenting them within the context of data synthesis, which reads "Recent advancements in hydrological modeling highlight the importance of combining ML approaches with diverse data sources and process-based models to improve prediction accuracy. Kratzert et al. (2021) demonstrated how

deep learning models could leverage the synergy among multiple meteorological datasets to enhance rainfall-runoff predictions, emphasizing the role of data integration in improving model performance. Similarly, Razavi et al. (2022) advocated for the coevolution of machine learning and process-based models, suggesting that their combined use can address limitations inherent to each approach and revolutionize Earth and environmental sciences. Reichstein et al. (2019) explored the intersection of data-driven methods and process understanding, illustrating how deep learning can advance Earth system science by extracting insights from complex datasets while maintaining a connection to fundamental physical principles. These studies underscore the critical interplay between information quantity, quality, and model design, which is central to this study."

**RC2.3**: Line 80: Can you discuss the rationale for choosing RF, SVM, and ANN as the machine learning models for streamflow predictions to test your hypothesis? Recent studies have suggested that Long Short-Term Memory (LSTM) networks are the state-of-the-art machine learning models for time-series river flow predictions, outperforming other approaches.

**Response to RC2.3**: We understand your point. However, the goal of this study is to investigate the sensitivity of prediction accuracy to the quantity and quality of input information, not to compare or identify the "best" model for streamflow prediction. RF, SVM, and ANN provide a range of modeling approaches with distinct characteristics, allowing us to explore how different model architectures handle varying training data scenarios. For example, RF is a robust ensemble-based model and is well-suited for examining feature importance and handling noise in datasets. SVM is effective for high-dimensional, particularly in identifying decision boundaries, making it useful for scenarios with limited training data. ANN serves as a baseline neural network model for testing sensitivity to diverse input data and can be extended to more complex architectures such as LSTM if needed. We acknowledge that LSTM networks have demonstrated state-of-the-art performance in time-series river flow predictions due to their ability to capture temporal dependencies. However, LSTMs, as a type of recurrent neural network, are particularly well-suited for tasks involving temporal or time-series analysis and other sequential data. In contrast, this study aims to derive generalizable insights across various model types. In addition, LSTMs, though powerful, require larger datasets and significant computational resources to optimize effectively, which may complicate isolating the effects of input information quality and quantity. RF, SVM, and ANN remain widely used in hydrological and environmental modeling, making our findings applicable to a broader audience and practical for researchers or practitioners working with simpler or more fundamental machine learning models. Incorporating LSTM models, while valuable, would narrow the focus to advanced neural network architectures, which is beyond the immediate scope of this study.

**RC2.4**: Table 1: In the current model setup, for both water quantity and water quality predictions, the baseline inputs (WDO) include only climate variables. However, in reality, nutrient inputs (e.g., fertilizers, human waste, and manure) are essential for predicting N and P, regardless of whether using machine learning or process-based models. Do you think it is fair to test TN and TP predictions when WDO includes only climate variables? If nutrient inputs were incorporated, adding loads from SWAT might have less impact. How much of the conclusions from this study can be applied to water quality ML studies where nutrient inputs are typically included as predictors?

**Response to RC2.4**: We appreciate your concerns regarding the ability to predict N and P using only meteorological data, and we agree that nutrient inputs (e.g., fertilizers, human waste, and manure) are critical for accurate predictions. However, it is important to note that the influence of weather data on watershed hydrological responses has been well-documented in previous studies (e.g., Chen et al., 2022; Rattan et al., 2019). Weather data play a significant role in predicting not only water quantity but also water quality. That said, the absence of key information in agricultural management practices and nutrient inputs, does contribute to reduced accuracy in predicting N and P in this study. However, the primary goal of our study is not to assess whether meteorological data alone can predict flow, SS, TN, or TP. Rather, we aim to explore how to efficiently enhance the accuracy of hydrological machine learning models by examining the quantity and quality of training data that drive improvements. In this context, the weather dataset serves as a baseline (WDO) to which other scenarios, including those incorporating uncalibrated and calibrated SWAT model outputs, are compared. This baseline allows us to evaluate the incremental value of additional data sources. Notably, the outputs of mechanistic modeling, which account for nutrient inputs, agricultural practices, and various transport processes, demonstrated a significant improvement in both the transfer entropy (information quality) to the target variables and the overall prediction accuracy. By leveraging mechanistic modeling outputs, we highlight how incorporating additional datasets with nutrient-related information can substantially enhance machine learning predictions, reinforcing the importance of such inputs in water quality modeling.

**References**

Chen, P., Li, W., He, K., 2022. Impacts of different types of El Niño events on water quality over the Corn Belt, United States. Hydrol. Earth Syst. Sci. 26, 4875-4892.

Rattan, K.J., Blukacz-Richards, E.A., Yates, A.G., Culp, J.M., Chambers, P.A., 2019. Hydrological variability affects particulate nitrogen and phosphorus in streams of the Northern Great Plains. Journal of Hydrology: Regional Studies 21, 110-125.

**RC2.5**: Line 255: More information about the SWAT model is needed.

• The manuscript states that SWAT can incorporate management practices, and two of the watersheds are heavily farmed and urbanized. Were inputs and parameters representing agricultural and human processes that significantly impact water quality (TN, TP, and SSD) included and calibrated? Please provide the inputs, parameters, and calibrated values.

• It is also unclear how many SWAT models were developed. Was a single SWAT model used per basin, or was a separate model created for each target variable within a basin?

• Related to (2), can you clarify the calibration process? Was a weighted multi-objective calibration approach applied, or were parameters calibrated for flow first, followed by calibration for water quality?

**Response to RC2.5**: Thank you for your comments and questions. We apologize for the insufficient explanation regarding the setup and calibration process of the SWAT model in the original manuscript. Agricultural management practices were incorporated based on the Agricultural Work Schedule Document from the Rural Development Administration of Korea (RDA, 2014). While we initially mentioned that "the agricultural management practices compiled from the study watersheds were incorporated into both models (RDA, 2014)," the details were not provided. In the revised manuscript, we have included a supplementary table (Table S2) detailing the application schedules and rates of management practices used in SWAT modeling. Then, the description was modified to "the agricultural management practices, including fertilizer application, planting and harvest dates, compiled from the study watersheds were incorporated into both models (RDA, 2014; Table S2)," in the revised manuscript.

The calibration of the SWAT model was conducted separately for each watershed, following a sequential approach where upstream watersheds were calibrated first, and their calibrated parameter values were not changed when calibrating the corresponding parameters for their downstream watersheds. For example, the WJ watershed (which is nested by the HN watershed) was calibrated prior to the HN watershed, and then the parameters for areas that are not included in the WJ watershed but only in the HN watershed were calibrated to observations made at the outlet of the HN watershed. Regarding the number of SWAT models, a single model was developed for each watershed, and each target variable (streamflow, SS, TN, and TP) was calibrated within that model. Calibration followed the widely recommended sequence (Arnold et al., 2012; Engel et al., 2007; Santhi et al., 2001), prioritizing streamflow calibration first, followed by SS, and finally TN and TP, to account for the interdependencies among these constituents due to shared transport processes.

To address your concern about calibration inputs and parameters, we have added a supplementary table (Table S3) listing the calibrated parameters, their values, and their corresponding hydrological responses for each watershed. Additionally, we revised the

subsection "2. Methods and Materials: 2.4 Theory-driven (or mechanistic) model" to provide a clearer explanation of the calibration process and parameter selection. Specifically, we now state in the revised manuscript:

"The parameter values of the SWAT models were calibrated to flow, SS, TN, and TP observations made at the study watersheds' outlets (Table S3). While the SWAT model includes many parameters, previous studies (Arnold et al., 2012; Douglas-Mankin et al., 2010; El-Sadek and Irvem, 2014) have identified key parameters with high sensitivity to each hydrological response. In this study, we focused on these parameters for each target variable and calibrated each watershed independently. Calibration was performed sequentially, upstream watersheds were calibrated first, and their calibrated parameter values were not changed when calibrating the corresponding parameters for their downstream watersheds. For example, the WJ watershed (which is nested by the HN watershed) was calibrated prior to the HN watershed, and then the parameters for areas that are not included in the WJ watershed but only in the HN watershed were calibrated to observations made at the outlet of the HN watershed. For target variables, streamflow was calibrated first, followed by SS, and then TN and TP, based on the interdependencies among these constituents resulting from shared transport processes."

We hope these revisions provide the necessary clarity and address your concerns comprehensively.

**Table S2.** Management practice application schedules and rates considered in the SWAT modeling.

| Month | Day | Operation | Value | Crop |
|-------|-----|-----------|-------|------|
| 5 | 1 | Fertilizer application | 67.5 kg ha-1 (N) 45.0 kg ha-1 (P) | |
| 6 | 1 | Plant/begin growing season | | |
| 7 | 10 | Fertilizer application | 22.5 kg ha-1 (N) | Rice |
| 10 | 1 | Harvest and kill operation | | |

**Table S3.** Calibrated parameters and their values in the SWAT model for each hydrological response in the study watersheds.

| Variables | Parameters | Calibrated Values | | | |
|-----------|-----------|-------|-------|-------|-------|
| | | WJ | HN | JS | PYJ |
| Flow | CN2.mgt | 58.0 | 51.4 | 81.5 | 55.0 |
| | ALPHA_BF.gw | 0.02 | 0.58 | 0.12 | 0.23 |
| | GW_DELAY.gw | 424.6 | 470.4 | 414.6 | 34.2 |

| | | | | |
|---|---|---|---|---|
| | GW_REVAP.gw | 0.12 | 0.15 | 0.09 | 0.12 |
| | GWQMN.gw | 3121 | 1671 | 404.2 | 125.0 |
| | EPCO.bsn | 0.20 | 0.55 | 0.33 | 0.19 |
| | ESCO.bsn | 0.57 | 0.98 | 0.01 | 0.39 |
| | SUR_LAG.bsn | 3.66 | 7.22 | 1.47 | 19.8 |
| | OV_N.hru | 0.23 | 0.23 | 0.28 | 0.14 |
| | CH_N2.rte | 0.15 | 0.18 | 0.17 | 0.27 |
| | CH_K2.rte | 85.3 | 5.42 | 21.3 | 0.83 |
| | SOL_AWC.sol | 0.04 | 0.09 | 0.03 | 0.51 |
| | USLE_P.mgt | 0.33 | 0.22 | 0.29 | 0.81 |
| | SPCON.bsn | 0.005 | 0.0004 | 0.008 | 0.003 |
| | SPEXP.bsn | 1.42 | 1.24 | 1.08 | 1.43 |
| | ADJ_PKR.bsn | 1.46 | 1.15 | 0.97 | 0.70 |
| | EROS_SPL.bsn | 1.08 | 3.00 | 1.40 | 2.14 |
| | EROS_EXPO.bsn | 2.43 | 1.94 | 1.78 | 2.08 |
| SS | C_FACTOR.bsn | 0.07 | 0.30 | 0.28 | 0.01 |
| | RILL_MULT.bsn | 0.57 | 0.98 | 0.54 | 1.97 |
| | PRF.bsn | 1.47 | 0.40 | 0.17 | 1.71 |
| | CH_D50.bsn | 84.5 | 60.3 | 45.0 | 54.1 |
| | USLE_K.sol | 0.64 | 0.65 | 0.47 | 0.65 |
| | CH_COV1.rte | 0.53 | 0.23 | 0.68 | 0.71 |
| | CH_COV2.rte | 0.64 | 0.31 | 0.65 | 0.36 |
| | USLE_C.dat | 0.49 | 0.40 | 0.34 | 0.42 |
| | BIOMIX.mgt | 0.13 | 0.68 | 0.15 | 0.67 |
| | LAT_ORGN.gw | 104.9 | 191.9 | 58.7 | 26.2 |
| | RCN.bsn | 3.68 | 4.85 | 11.0 | 9.68 |
| | N_UPDIS.bsn | 12.4 | 17.0 | 74.8 | 85.3 |
| | NPERCO.bsn | 0.59 | 0.17 | 0.86 | 0.29 |
| TN | CMN.bsn | 0.003 | 0.003 | 0.002 | 0.002 |
| | CDN.bsn | 2.05 | 0.60 | 2.66 | 0.60 |
| | SDNCO.bsn | 0.35 | 0.94 | 0.76 | 0.16 |
| | ERORGN.hru | 0.41 | 4.91 | 0.13 | 0.25 |
| | RS4.swq | 0.04 | 0.01 | 0.09 | 0.05 |
| | BC3.swq | 0.35 | 0.25 | 0.22 | 0.35 |
| | LAT_ORGP.gw | 33.1 | 8.50 | 2.10 | 1.00 |
| | P_UPDIS.bsn | 30.1 | 30.0 | 6.55 | 0.90 |
| | PPERCO.bsn | 15.1 | 17.1 | 16.3 | 12.5 |
| TP | PSP.bsn | 0.56 | 0.61 | 0.36 | 0.69 |
| | ERORGP.hru | 0.03 | 0.89 | 0.05 | 0.48 |
| | RS5.swq | 0.04 | 0.04 | 0.09 | 0.03 |
| | BC4.swq | 0.46 | 0.49 | 0.30 | 0.11 |

**References**

Arnold, J.G., Moriasi, D.N., Gassman, P.W., Abbaspour, K.C., White, M.J., Srinivasan, R., Santhi, C., Harmel, R., Van Griensven, A., Van Liew, M.W., 2012. SWAT: Model use, calibration, and validation. Transactions of the ASABE 55, 1491-1508.

Engel, B., Storm, D., White, M., Arnold, J., Arabi, M., 2007. A hydrologic/water quality model application. JAWRA Journal of the American Water Resources Association 43, 1223-1236.

Rural Development Administration (RDA), 2014. Agricultural work schedule – Machine transplanting cultivation. http://www.nongsaro.go.kr.

Santhi, C., Arnold, J.G., Williams, J.R., Dugas, W.A., Srinivasan, R., Hauck, L.M., 2001. Validation of the swat model on a large river basin with point and nonpoint sources. JAWRA Journal of the American Water Resources Association 37, 1169-1188.

**RC2.6**: Figure 4: 3D plots are often hard to interpret. Consider using 2D plots for model performance comparison.

**Response to RC2.6**: Thank you for the suggestion; we understand your concerns and have made the necessary adjustments. The figures have been revised to a 2D format for improved clarity, with each watershed represented by a distinct symbol. Additionally, the training dataset is now plotted along the x-axis for better visualization.

[Figure]

(a) Flow

[Figure]

**Figure 4.** Prediction accuracy (KGE) of hydrological ML models trained with the different training data set combinations. The KGE values that do not satisfy the acceptable accuracy levels (e.g., i.e., 0.54 for flow, 0.17 for SS, and -0.03 for TN/TP) are included in gray areas.

**RC2.7**: Figure 5: Could you adjust the figure size to make the duration curves less flat? It's difficult to distinguish the differences. What is the scale of the Y-axis? It appears to be on a log scale.

**Response to RC2.7**: Thank you for the suggestion on Figure 5. We have adjusted the figure size, specifically focusing on the Y-axis, to make the duration curves less flat and easier to interpret. The Y-axis is indeed presented on a log scale in the original figure to emphasize differences at both ends of the exceedance probabilities (small and large). To further improve clarity, we have added additional plots with both normal (linear) and log-scale Y-axes (Figures 5 and S5) in the supplementary document. This allows readers to better understand the differences across all flow ranges, including extreme values (small and large exceedance probabilities) as well as intermediate probabilities, such as 50%. We hope these adjustments address your concerns and enhance the clarity of the presented data.

[Figure]

(a) the WDO and WD+UC cases.

[Figure]

(b) the WD+C and All cases.

**Figure 5.** Comparison between observed and ML-predicted FDCs at the outlet of the WJ watershed.

[Figure]

(a) the WDO and WD+UC cases.

[Figure]

(b) the WD+C and All cases.

**Figure S5.** Comparison between observed and ML-predicted FDCs in normal scale at the outlet of the WJ watershed.

**RC2.8**: Figure 6: All basins and target variables show a similar pattern, with the key takeaway being that ME increases with the amount of data, regardless of data correlation. You might consider leaving one subfigure and moving the rest to the supplemental section, or consolidating them into a single figure. For example, as the conceptual figure suggests, the X-axis could represent different data, the Y-axis ME, and different lines could indicate target variables. Adding uncertainty bands could also capture watershed variance. Just some suggestions to consider.

**Response to RC2.8**: We agree with your suggestion that Figure 6 shows similar trends across variables and that consolidating them into a single figure would be beneficial. The increases in ME due to the addition of training datasets for four watersheds and four target variables have been revised into a single figure. The Y-axis represents the percentage increase, with the WDO training dataset serving as the baseline, while the X-axis represents the target variables and training datasets, with each watershed represented by a distinct patterns. We believe that including all four watershed cases in the plot will highlight variations or uncertainties across the watersheds. In addition, we also added a graph of the ME values for each training dataset by watersheds with different target variables to the supplementary material, to enhance the clarity of the information amounts across the training datasets.

[Figure]

**Figure 6.** Increases in ME due to the addition of additional training data sets. The WDO training data set serves as the baseline for this comparison.

[Figure]

**Figure S8.** ME of the training data sets by the watersheds with different target variables.

**RC2.9**: Figure 7: This figure now has too many dimensions—different basins, target variables, model inputs, and ML models—which makes it quite confusing. Could you simplify it to highlight key finds, perhaps by leveraging the suggestions for Figure 6.

**Response to RC2.9**: We agree with your recommendation. We revised Figure 7 based on your suggestion for Figure 6. As transfer entropy values vary depending on the machine learning models, it was difficult to represent all four watershed cases in a single graph. Thus, we left only the graph for the WJ watershed (Figure 7) and moved the remaining watersheds (HN, JS, and PYJ) to the supplementary materials (Figure S9). In addition, we also added a graph of the TE values across the different basins, target variables, datasets, and machine learning models to the supplementary material, to enhance the clarity of the information quality across the training datasets (Figure S10).

[Figure]

**Figure 7.** Increases in TE due to the addition of training data sets from the WJ watershed. The WDO training data set serves as the baseline for this comparison.

[Figure]

(a) HN watershed

(b) JS watershed

[Figure]

(C) PYJ watershed

**Figure S9.** Increases in TE due to the addition of training data sets from different watersheds (HN, JS, and PYJ). The WDO training data set serves as the baseline for this comparison.

[Figure]

(a) WJ watershed

(b) HN watershed

(c) JS watershed

(d) PYJ watershed

**Figure S10.** TE of the training data sets by the machine learning algorithms with different target variables.

**RC2.10**: Figure 10: How well are the regressions? Can you report R2 and p-value?

**Response to RC2.10**: We plotted the responses of average KGE values against the ME of target variables to illustrate the general trend or relationship between them, rather than to develop a regression model for predicting KGE from ME. For your reference, we have provided the relevant statistics below (Tables RC2.10.A and RC2.10.B). It is important to note that these lines represent the overall relationship and should not be interpreted as having predictive capabilities. For clarification, we have revised the plot title to: "General trends in the average KGE scores of three ML models trained with four training data sets, in relation to the ME of the target variables."

[Figure]

**Figure 10.** General trends in the average KGE scores of three ML models trained with four training data sets, in relation to the ME of the target variables.

**Table RC2.10.A.** Summary statistics of coefficient of determination ($R^2$) and statistical significance (p-value) for the target variables.

| Statistics | Flow | SS | TN | TP |
|---|---|---|---|---|
| $R^2$ | 0.383 | 0.887 | 0.992 | 0.315 |
| P-value | 0.381 | 0.058 | 0.004 | 0.439 |

**Table RC2.10.B.** Summary statistics of ME of the target variables and averaged KGE scores of three ML models trained using four training data sets.

| Watershed | Flow | | SS | | TN | | TP | |
|---|---|---|---|---|---|---|---|---|
| | ME | KGE | ME | KGE | ME | KGE | ME | KGE |
| WJ | 8.680 | 0.675 | 5.507 | 0.490 | 6.080 | 0.408 | 5.757 | 0.475 |
| HN | 8.868 | 0.735 | 5.252 | 0.463 | 5.946 | 0.321 | 5.828 | 0.350 |
| JS | 7.896 | 0.681 | 4.014 | 0.109 | 5.924 | 0.305 | 5.760 | 0.254 |
| PYJ | 8.884 | 0.805 | 5.801 | 0.446 | 6.770 | 0.679 | 6.578 | 0.486 |

**RC2.11**: Discussion: Several points discussed are common knowledge, making them less novel and somewhat irrelevant. For example: (1) ML model accuracy depends on study watersheds, target variables, and ML model types; and (2) water quality is generally harder to predict than water quantity. The key findings of this work is its quantitative evaluation of data quality and quantity and their relationship to model performance. Could you expand on how these findings can guide future modeling efforts, such as optimizing input selection, implementing quality control measures, or integrating insights with process - based models.

**Response to RC2.11**: We appreciate your point and are grateful for your constructive and valuable suggestions. While it is widely recognized that ML model accuracy depends on factors such as study watersheds, target variables, and model types, these observations were included to establish context and provide a baseline for interpreting our findings. For example, discussing the challenges of predicting water quality versus water quantity helps frame the importance of our quantitative evaluation of input data characteristics and their effects on the prediction accuracy of water quantity and quality. To address the suggestions, we added additional discussion, which now reads, "The quantitative evaluation of data quality and quantity in relation to model performance provides actionable insights for future modeling efforts. Our findings highlight the importance of prioritizing high-quality, high-relevance inputs to improve prediction accuracy. Future studies can focus on including inputs that have strong statistical relationships to target variables. Feature selection techniques, informed by our results, could help identify the most impactful variables, reducing the risk of overfitting and computational inefficiencies. The demonstrated sensitivity of ML models to data quality suggests the need for rigorous preprocessing steps, such as outlier detection, imputation of missing data, and validation of sensor measurements. For example, ensuring consistent and accurate data collection at critical watershed locations can enhance model reliability. Our results could guide the development of benchmarks or thresholds for data quality metrics (e.g., measurement error limits) to ensure datasets meet the minimum requirements for effective modeling.

Out findings also have implications for hybrid modeling approaches, particularly guiding the integration of ML with process-based models. These results can help select variables where process-based models provide better physical realism (e.g., water quality dynamics) while leveraging ML for complementary predictions (e.g., extreme event responses or data gap interpolation). Improving the quality of key inputs can also help reconcile discrepancies between data-driven and process-based predictions, enhancing overall model accuracy. Expanding beyond standalone ML applications, this study provides a foundation for adaptive modeling frameworks that dynamically assess and adjust data inputs based on their predictive contributions.

Future studies could build on our quantitative evaluation by applying these insights to develop guidelines or workflows for selecting and preparing datasets tailored to specific hydrological and water quality objectives. The study underscores the need to refine hydrological ML prediction models by emphasizing the connection between data quality and prediction accuracy. Incorporating high-quality training data into ML training can significantly enhance the reliability

and efficiency of ML models. The integration of theory-driven and data-driven approaches will not only improve prediction accuracy but also streamline model training by ensuring that the data used contains both sufficient quantity and quality. Moreover, a deeper understanding of the interaction between different data types will inform more effective training strategies for ML models, leading to more accurate and reliable hydrological predictions."

**RC2.12**: Maybe I missed it. Did you talk about uncertainty and limitations of your work?

**Response to RC2.12**: Thank you for your constructive suggestion. In response to your comment, we have added a discussion on the limitations and uncertainties of this study, which reads "While this study provides valuable insights into the relationship between ML prediction accuracy and the quality and quantity of input data in hydrological modeling, several limitations and uncertainties must be acknowledged. The study's findings are based on a specific set of watersheds with unique hydrological, climatic, and geographical characteristics. The variability in watershed conditions across different regions may limit the generalizability of the results. The study assumes that the available datasets accurately represent real-world hydrological processes. However, biases, errors, and inconsistencies in the input data, such as measurement inaccuracies or missing values, could influence the results. The study evaluates model performance at specific temporal and spatial scales. Fine temporal resolutions (e.g., daily predictions) may introduce additional complexities not captured in coarser scales (e.g., monthly or annual). Despite optimization efforts, ML models remain susceptible to overfitting, particularly when trained on small datasets or when irrelevant features are included. By acknowledging these limitations and uncertainties, this study provides a foundation for future work to refine and expand upon its findings, ultimately improving the reliability of ML models in hydrological applications."